# Non-locality of the Earth's quasi-parallel bow shock: injection of thermal protons in a hybrid-Vlasov simulation

Markus Battarbee[1], Urs Ganse[1], Yann Pfau-Kempf[1], Lucile Turc[1], Thiago Brito[1], Maxime Grandin[1], Tuomas Koskela[1,2], and Minna Palmroth[1,3]

[1]Department of Physics, University of Helsinki, Helsinki, Finland
[2]Department of Physics and Astronomy, University of Turku, Turku, Finland
[3]Finnish Meteorological Institute, Helsinki, Finland

**Correspondence:** Markus Battarbee (markus.battarbee@helsinki.fi)

**Abstract.** We study the interaction of solar wind protons with the Earth's quasi-parallel bow shock using a hybrid-Vlasov simulation. We employ the global hybrid model Vlasiator to include effects due to bow shock curvature, tenuous upstream populations, and foreshock waves. We investigate the uncertainty of the position of the quasi-parallel bow shock as a function of several plasma properties, and find that regions of non-locality or uncertainty of the shock position form and propagate away from the shock nose. Our results support the notion of upstream structures causing patchwork reconstruction of the quasi-parallel shock front in a non-uniform manner. We propose a novel method for spacecraft data to be used to analyze this quasi-parallel reformation.

We combine our hybrid-Vlasov results with test-particle studies and show that proton energization, which is required for injection, takes place throughout a larger shock transition zone. Energization of particles is found regardless of the instantaneous non-locality of the shock front, in agreement with it taking place over a larger region. Distortion of magnetic fields in front of and at the shock is shown to have a significant effect on proton injection.

We additionally show that the density of suprathermal reflected particles upstream of the shock may not be a useful metric for the probability of injection at the shock, as foreshock dynamics and particle trapping appear to have a significant effect on energetic particle accumulation at a given position in space. Our results have implications for statistical and spacecraft studies of the shock injection problem.

## 1  Introduction

Collisionless plasma shocks are an ubiquitous source of plasma acceleration, common within stellar, planetary, and interplanetary environments. Shock dynamics have been studied in great detail at the Earth's bow shock. In regions of shock geometry where the angle $\theta_{Bn}$ between the shock-normal direction $\hat{n}$ and the upstream interplanetary magnetic field (IMF) direction **B** is small ($\lesssim 45°$), the shock is considered quasi-parallel (see, e.g., Burgess et al., 2005). In this region, if the shock is a

strong fast-mode supercritical shock, a fraction of thermal incident ions are reflected, streaming away from the shock along the magnetic field lines, forming the foreshock region (Fairfield, 1969; Eastwood et al., 2005). The streaming energized particles excite instabilities such as a right-hand ion-ion beam instability, building a wave field of ultra-low frequency (ULF) waves (Hoppe et al., 1981) with periods around $\sim 30\,\mathrm{s}$, which further interact with the particles themselves and are convected toward the bow shock. As the waves are convected with the supersonic solar wind flow, they appear mostly left-handed in the spacecraft frame. The incident ULF waves can experience nonlinear steepening, possibly forming shocklets (Hada et al., 1987; Wilson III, 2016) or short large amplitude magnetic structures (SLAMS; Schwartz et al., 1992; Burgess, 1995; Lucek et al., 2008), eventually causing patchwork reformation of the bow shock (Scholer and Terasawa, 1990; Thomas and Winske, 1990; Schwartz and Burgess, 1991; Burgess, 1995) as incoming structures proceed to build a new shock front periodically (Burgess, 1989). At Mercury this reformation has been studied through mainly magnetic field measurements in Sundberg et al. (2013). The complicated structure of the shock-associated transition region was linked with local reconnection in Gingell et al. (2019). As the location of the shock front is challenging to define due to movement i.e. nonstationarity of a well-defined shock front, the formation and convection of a new shock front, and even the disappearance of the old front, we now discuss this uncertainty of the shock position which we designate the "non-locality" of the shock. As plasma parameters across a quasi-parallel shock can be non-monotonic, non-locality encompasses more than mere thickness of a well-defined shock front. Our definition of non-locality can also be measured using spacecraft, providing a novel metric for quantifying space plasma observations. In this study, we limit our analysis to ion scales and assume the reformation of the quasi-parallel bow shock to happen on temporal and spatial scales similar to those of steepened ULF waves and associated transient structures.

An important open question for space physics and particle acceleration is the shock injection problem (see, e.g., Zank et al., 2001), or how exactly thermal particles are reflected at a super-critical quasi-parallel shock. Injection from a thermal population is a necessary step in efficient diffusive shock acceleration (DSA; Axford et al., 1977; Blandford and Ostriker, 1978; Bell, 1978; Krymsky et al., 1979), which is a major source of energetic particles throughout the universe. The injection problem has been studied extensively during the past decades with, amongst others, observations (Sckopke et al., 1983; Thomsen et al., 1983; Gosling et al., 1989; Johlander et al., 2016), analytical work (Schwartz et al., 1983; Malkov et al., 2016), test-particle modeling (Gedalin, 2001; Battarbee et al., 2011; Gedalin, 2016; Johlander et al., 2016), and particle-in-cell simulations (Caprioli et al., 2015; Liseykina et al., 2015; Sundberg et al., 2016; Hao et al., 2016; Caprioli et al., 2017). Significant historical work using 1-D or 2-D local hybrid simulations can be found in, e.g., Burgess (1989); Scholer (1990) and Kucharek and Scholer (1991). Previous studies have suggested three methods for injection: Specular reflection (Gosling et al., 1982), shock drift acceleration (SDA; Giacalone, 1992; Lever et al., 2001; Burgess, 1987) and associated shock surfing (Lever et al., 2001), and thermal leakage from the downstream (Ellison, 1981; Edmiston et al., 1982; Lyu and Kan, 1990; Malkov, 1998). These three methods were derived from assumptions of macroscopic, planar, and stationary shock fronts and are thus limited, but an important first step towards understanding the concept. Magnetic mirroring as described through quasi-linear theory and conservation of the first adiabatic invariant is usually excluded, as changes to magnetic fields may occur on scales much smaller and faster than those of ion gyromotion.

In this paper, we investigate the complex structure and non-locality of the Earth's quasi-parallel bow shock as well as the injection problem both through hybrid-Vlasov simulations and test-particle runs. In section 2 we present our hybrid-Vlasov simulations. In section 3 we present results from two different hybrid-Vlasov datasets. In section 4 we introduce our test-particle simulation method, and in section 5 we present results of test-particle injection and energization. Section 6 presents analysis and discussion on our findings, and we present our conclusions in section 7.

Throughout this study, we use the following terminology:

- An *injected* particle has interacted with the bow shock and returned to the upstream. This may also be called *reflection*. During this process, particles gain energy in the solar wind frame.

- A *transmitted* particle has passed through the bow shock to the far downstream. The particle may or may not be energized during this process.

- *Energization* is when during a single shock encounter, a particle gains energy in the solar wind frame so that it is no longer part of the incident plasma thermal distribution.

- *Acceleration* is when injected particles continue to gain energy through continuous and/or repeated shock interactions, such as DSA. This takes place over longer temporal and spatial scales, and is outside the scope of this study.

- *Non-locality* of the quasi-parallel bow shock is a measure of the disagreement between different measurements of where the bow shock is locally estimated to be. This could also be referred to as the uncertainty of the shock position.

- The *shock-normal direction* $\hat{\mathbf{n}}$ is normal to the local, reforming shock front. This direction is highly variable.

- The *bow-normal direction* $\hat{\mathbf{n}}'$ is the normal direction for a parabola, estimating the global shape of the shock front. This direction is very stable.

- The *shock-normal angle* $\theta_{Bn}$ is the angle between the upstream magnetic field and the shock-normal direction. The shock-normal direction or a vector antiparallel to it is chosen in order to constrain the value to $\theta_{Bn} \in [0°, 90°]$. Due to fluctuations of both the upstream field and the local shock front, this angle is very unpredictable.

- The *bow-normal angle* $\theta_{Bn'}$ is the angle between the upstream magnetic field and the bow-normal direction. Like $\theta_{Bn}$, it is usually limited to $\theta_{Bn'} \in [0°, 90°]$, but in regions of significant mangetic field deformation, is allowed to have values $> 90°$. This measure allows analysis of shock interaction due to upstream magnetic field fluctuations while smoothing out the local reformation effects of the quasi-parallel shock front.

## 2  Vlasiator simulation

In modeling the Earth's bow shock, we employ Vlasiator (von Alfthan et al., 2014; Pfau-Kempf, 2016; Palmroth et al., 2018), a hybrid-Vlasov code designed to simulate the Earth's magnetosphere and the surrounding space environment. Vlasiator mod-

els kinetic proton-scale plasma physics by calculating the evolution of the proton distribution function on a Cartesian 3-dimensional velocity grid within each cell of a Cartesian spatial grid. In the presented runs, the spatial simulation domain is 2-dimensional. Modeling distribution functions directly instead of using a particle-in-cell method allows for accurate analysis of even the tenuous portions of non-thermal populations in the foreshock, and gives us a realistic model of foreshock and bow shock evolution. The noise-free distribution function formalism further allows using the magnetic field $\mathbf{B}$ and electric field $\mathbf{E}$ values as input to test-particle studies without a need for low-pass filtering.

Vlasiator models ions as distribution functions, solving the Vlasov equation for the ion (proton) distribution with electrons modeled as a cold massless charge-neutralizing fluid. Closure is provided via Ohm's law, including the Hall term. We assume that effects due to the electron pressure gradient can be neglected. Vlasiator is capable of modeling a number of ion kinetic effects even without resolving the ion skin depth (Pfau-Kempf et al., 2018), and Hoilijoki et al. (2017) reported how Vlasiator simulated global reconnection rates in agreement with empirical formulae. Dubart et al. (2020) shows that resolving the ion inertial length is not required for correctly resolving EMIC waves in the magnetosheath. Vlasiator has been used for several interesting foreshock and bow shock studies (Palmroth et al., 2015; Pfau-Kempf et al., 2016; Turc et al., 2018; Blanco-Cano et al., 2018; Turc et al., 2019). Our choice of simulation parameters do not quite resolve the ion inertial length, but instead ensure correct scale separation between global and local dynamics (e.g. bow shock curvature and ULF-wave induced shock ripples) and a noise-free representation of both thermal and non-thermal plasma. Tóth et al. (2017) have investigated how reconnection physics were affected by overresolving the inertial length (at the expense of scale separation), but they did not study the consequences of underresolving it.

In this paper, we use two datasets (simulations S1 and S2) modeling two different bow shock strengths and interplanetary magnetic field intensities. Results from these simulations have previously been published in Palmroth et al. (2015), Turc et al. (2018), and Turc et al. (2019). They are ecliptic plane ($x - y$) 2D–3V simulations (2D in the spatial domain, 3D in the velocity domain) parametrized using the Geocentric Solar Ecliptic (GSE) coordinate system with no tilt for the Earth's dipole. The $x$-coordinate is along the Earth-Sun axis, the $z$-coordinate is aligned with the Earth's magnetic axis, and the $y$-coordinate completes the right-handed system. We save variables such as field values and distribution function moments every $0.5\,\mathrm{s}$. The simulation extent is $2000 \times 1750$ spatial cells, covering the ranges $x \in [-7.7, 63.6]\,r_\mathrm{E}$ and $y \in [-31.3, 31.3]\,r_\mathrm{E}$ where $r_\mathrm{E} = 6371\,\mathrm{km}$ is the Earth radius. The simulation domain extent in the $z$-direction is only one cell thick with periodic boundary conditions. Each spatial cell is a cube of length $228\,\mathrm{km}$ along each edge. Our velocity domain employs a sparse representation (von Alfthan et al., 2014) and has a resolution of $30\,\mathrm{km\,s^{-1}}$. The simulation domain is initialized with a somewhat fast and hot solar wind inflow of $n_\mathrm{p,sw} = 3.3 \times 10^6\,\mathrm{m^{-3}}$, $T = 0.5\,\mathrm{MK}$, $\mathbf{V} = (-600, 0, 0)\,\mathrm{km/s}$. The magnetic field in simulation S1 is $\mathbf{B}(\mathrm{S1}) = (-5\cos 5°, 5\sin 5°, 0)\,\mathrm{nT}$ whereas in simulation S2 it is $\mathbf{B}(\mathrm{S2}) = 2\mathbf{B}(\mathrm{S1}) = (-10\cos 5°, 10\sin 5°, 0)\,\mathrm{nT}$. The quasi-radial IMF in these runs allows us to focus on the quasi-parallel bow shock. The somewhat hot solar wind ensures the inflow Maxwellian distribution is resolved adequately. The Earth's magnetic dipole is implemented at a realistic value of $8.0 \times 10^{22}\,\mathrm{Am^2}$, and the simulation domain inner boundary is a perfectly conducting sphere located at $r = 31800\,\mathrm{km}$ or about $5\,r_\mathrm{E}$. The simulation set-up results in solar wind Alfvénic Mach numbers of $M_\mathrm{A,1} \sim 10$ and $M_\mathrm{A,2} \sim 5$ and magnetosonic Mach numbers of $M_\mathrm{ms,1} \sim 5.4$ and $M_\mathrm{ms,2} \sim 3.8$ in front of the bow shock nose, and thus, strong fast-mode supercritical shocks. The

simulations were run for $t_{\mathrm{max},1} = 685\,\mathrm{s}$ and $t_{\mathrm{max},2} = 539\,\mathrm{s}$, respectively. To facilitate comparison with existing numerical stud-

ies, we note that for both simulation runs the solar wind ion inertial length is $125.4\,\mathrm{km} = 0.020\,r_{\mathrm{E}}$, and for S1, the solar wind

plasma beta $\beta_1 = 2.3$, and for S2, $\beta_2 = 0.57$

Figure 1 depicts the Vlasiator simulation domain for simulation S1. The color map depicts proton densities, showing a dense

magnetosheath between the bow shock and the magnetosphere, as well as variations in the upstream plasma density within

the proton foreshock region. A fuchsia contour depicts where plasma density has increased two-fold over solar wind values,

providing a rough estimate of the bow shock position. Black lines illustrate magnetic field lines, showing how the foreshock

is permeated by fluctuations, as well as visualizing the complicated nature of magnetic flux compression and deflection at

the quasi-parallel bow shock. The white circle indicates the simulation inner boundary, and two overlapping white rectangles

indicate our regions of interest within the simulation. The larger white rectangle is used for visualizing test-particle studies of

proton injection, whereas the smaller rectangle is used for analysis of quasi-parallel bow shock non-locality. Plotting artefacts

for magnetic field lines at $X < 10\,r_{\mathrm{E}}$ are a result of visual post-processing and are not present in the scientific results presented.

## 3    Vlasiator results

In this section, we present results of hybrid-Vlasov simulations. First, we fit the global position of the bow shock using a

quartic estimation and calculate the bow-normal angle to estimate the general direction of the shock normal. As our fit is so

close to a parabola, we will henceforth for simplicity refer to it as a parabola. Then, we use several local measurements of

plasma properties to estimate the rapidly moving and varying local position of the shock, and use their disagreement to define

a non-locality of the shock.

### 3.1    Bow shock location and the shock-normal angle

In previous hybrid-method investigations into ion injection at kinetic plasma shocks, the shock descriptions have been usually

either 1-D (see, e.g., Lyu and Kan, 1990; Scholer, 1990; Scholer and Terasawa, 1990; Onsager et al., 1991; Su et al., 2012)

or if 2-D or 3-D, limited to local geometries (Guo and Giacalone, 2013; Caprioli et al., 2015; Hao et al., 2016; Sundberg

et al., 2016; Caprioli et al., 2017). In a local planar shock, it is feasible to simply define the shock-normal direction from

simulation box parameters and evaluate 1-D cuts along this line for defining the shock shape. However, as seen in Figure 1,

in a global 2-D simulation, the curved bow shock has a bow-normal direction dependent on the nose angle $\phi = \arctan(y/x)$,

which complicates evaluating the shock-normal direction (Thomas and Winske, 1990). Shock and injection investigations

within global simulations have recently been published in, e.g., Savoini et al. (2010, 2013); Karimabadi et al. (2014); Savoini

and Lembège (2015).

We now determine a rough estimate of the global bow shock shape. We do this by finding the contour where plasma density

increases two-fold over the solar wind value ($n_{\mathrm{p}} > 2n_{\mathrm{p,sw}}$). The value of $2n_{\mathrm{p,sw}}$ was chosen based on visual inspection. We

then fit a 4th order polynomial

$$r_s(\phi) = a_0 + a_1\phi + a_2\phi^2 + a_3\phi^3 + a_4\phi^4 \tag{1}$$

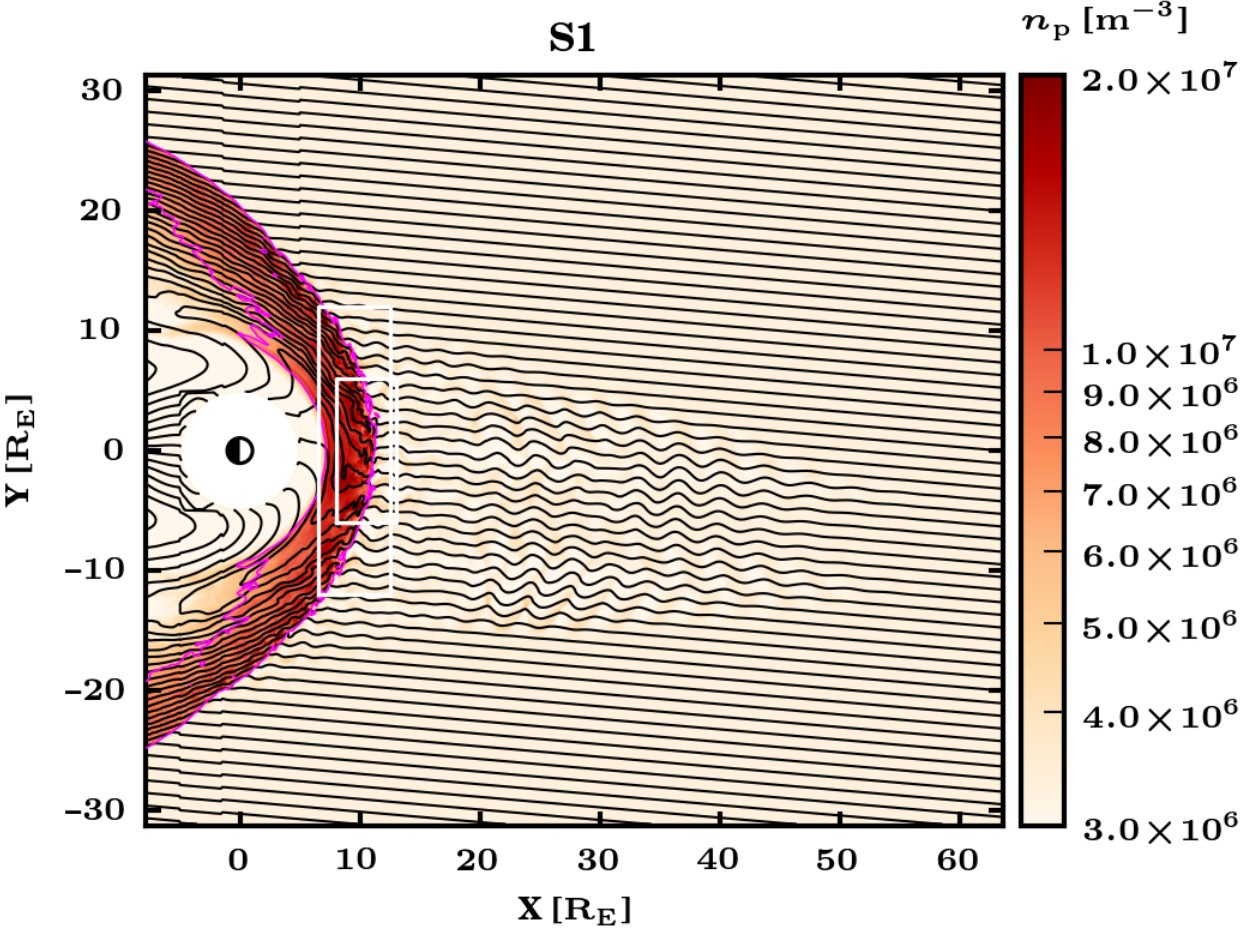

**Figure 1.** Overview of the Vlasiator simulation S1 ($B_{\mathrm{IMF}} = 5\,\mathrm{nT}$, $M_{\mathrm{A}} = 10$) at time $t = 500\,\mathrm{s}$, with proton number density (colormap) overlaid with an estimate of the bow shock position according to plasma compression (fuchsia curve, $n_{\mathrm{p}} > 2n_{\mathrm{p,sw}}$). Also shown are magnetic field lines (black curves) and two white overlapping rectangles indicating zoom-in regions used for analysis of local bow shock structure (smaller rectangle) and test-particle studies (larger rectangle).

using the nose angle and the radial distance $r = \sqrt{y^2 + x^2}$ at each contour position. This fit is performed at times $t_0 = 438\,\mathrm{s}$ and $t_f = 538\,\mathrm{s}$. We found that intermediate time steps are described well by performing linear interpolation in time of the polynomial coefficients.

One of the most commonly used criteria for defining the dynamics and injection characteristics of a shock is the shock-normal angle $\theta_{Bn}$, i.e., the angle between the shock-normal direction and the upstream magnetic field. The upstream magnetic field direction in the quasi-parallel shock region varies greatly due to upstream fluctuations (Greenstadt and Mellott, 1985). Thus, even within the quasi-parallel regime, the shock may exhibit a wide variety of shock-normal angles.

     As the shock front evolves, reforms, and fluctuates, the local shock-normal direction also evolves. The local instantaneous
shock-normal direction can end up being perpendicular or even reversed to the mean bow shock direction, and is thus challenging to evaluate in a meaningful manner. In this study, we define an alternative measure, the *bow-normal direction* $\hat{\mathbf{n}}'$, which is the normal direction for the parabolic fit to the mean shape of the global shape of the shock front. This is calculated as

$$\mathbf{n}' = (-\frac{\mathrm{d}r(\phi)}{\mathrm{d}\phi}\cos\phi + r(\phi)\sin\phi, \frac{\mathrm{d}r(\phi)}{\mathrm{d}\phi}\sin\phi + r(\phi)\cos\phi, 0) \tag{2}$$

and accordingly $\hat{\mathbf{n}}' = \mathbf{n}'/n'$. We use this bow-normal direction both for defining the bow-normal plasma bulk velocity compo-
nent, used for calculating the magnetosonic Mach number of the shock, and for defining a bow-normal angle $\theta_{Bn'}$, describing the angle between the local wave-distorted magnetic field and the bow-normal direction.

## 3.2    Shock non-locality

The locations of quasi-perpendicular and subcritical collisionless plasma shocks can, for the most part, be estimated well due to the upstream remaining undisturbed. However, at supercritical quasi-parallel shocks, the upstream is characterized by
magnetic and density fluctuations and an abundance of suprathermal particles. This can make defining the exact position of the quasi-parallel shock challenging. This localization is further hindered by the fact that the position of the shock changes locally at timescales related to shock reformation. Additionally, the global position of the shock changes at larger timescales due to variation in solar wind driving conditions. This non-stationarity of the shock is observed as, e.g., spacecraft encountering the shock multiple times during what is expected to be a single crossing (see, e.g., Lucek et al., 2002; Sundberg et al., 2016; Gingell
et al., 2017). In order to investigate the injection problem, we now attempt to define the local quasi-parallel shock position within a larger shock transition zone (Burgess, 1995) on reformation-related timescales. We also present a novel method for quantifying the uncertainty of the shock position, suitable for use in spacecraft observations and future investigations of the quasi-parallel bow shock.

     We evaluate the location of the shock as a transition between the upstream and downstream conditions using three plasma
properties. The first is plasma compression, using the previously introduced criterion of $n_{\mathrm{p}} > 2n_{\mathrm{p,sw}}$. The second is heating of the solar wind core population, $T_{\mathrm{core}} > 4T_{\mathrm{sw}}$, similar to the method of Wilson III et al. (2014b, a), with the value $4T_{\mathrm{sw}}$ selected based on visual inspection. To achieve this, the Vlasiator distribution function is split into core and suprathermal parts ($n_{\mathrm{p,core}}$ and $n_{\mathrm{p,st}}$). The plasma contained in each velocity space cell is evaluated as belonging to the core distribution if it is inside a sphere centered at $u_{\mathrm{sw}} = (-600, 0, 0)\,\mathrm{km\,s^{-1}}$ and with a radius of $690\,\mathrm{km\,s^{-1}}$. Cells outside this sphere are considered

as belonging to the suprathermal distribution. The third criterion is when the plasma magnetosonic Mach number, calculated using the local fast magnetosonic mode speed and the bow-normal plasma bulk velocity, falls below 1. We do not include any further criteria based on the magnetic field direction or magnitude, as magnetic field compression at a quasi-parallel shock is sporadic and limited, and the transition region has a wide range of local field orientations (see, e.g., Figure 1 of Gingell et al., 2019). We emphasize that the presented methods will potentially register shocklets and SLAMS as they take part in the reformation process.

In Figure 2 we present in panels (a) and (b) snapshots of plasma density from simulations S1 and S2, respectively, at time $t = 500\,\mathrm{s}$, zoomed in on the nose of the quasi-parallel bow shock (indicated by the smaller white rectangle in Figure 1). We have plotted the plasma density with overlaid contours representing the bow shock positions according to criteria for plasma density (fuchsia), solar wind core heating (green), and magnetosonic Mach number (pale blue).

The three contours are highly variable and agree on the position of the quasi-parallel shock only on the order of 50% of the time. We have selected four positions for profile cuts, depicted by black dashed lines in panel (a), showcasing different kinds of shock crossings. These simulate what a spacecraft might observe, except that they are spatial instead of temporal profiles. Line profiles for the three plasma properties used to gauge the shock position are shown in panels (c), (d), (e), and (f). Graphed quantities are scaled so that a value of 1 is where the shock is estimated to be. The distance between the positions of bow shock parametrization closest and farthest from the Earth is the disagreement between the three parametrizations, and is shown as shaded gray regions. This distance estimates the uncertainty of the shock position, or the extent of the shock transition region within which the three plasma properties estimate the shock to be. We designate this distance the shock *non-locality*. It is defined in units of Earth radii instead of, e.g., upstream gyroscales in order to facilitate comparison of bow shock structure sizes between different IMF conditions.

The cut shown in panel (c), at $Y = 3.8\,r_\mathrm{E}$, shows regions of low plasma density in what would appear to be the downstream, likely a result of a new shock front forming at $X \approx 11\,r_\mathrm{E}$, with the old shock position closer to $X \approx 10.5\,r_\mathrm{E}$. Panel (d), at $Y = 2.8\,r_\mathrm{E}$, shows active reformation of the quasi-parallel bow shock, with the first and last estimated shock positions disagreeing by over $1.0\,r_\mathrm{E}$, as a new front is forming at $X \approx 11.7\,r_\mathrm{E}$. The cut in panel (e), at $Y = 1.2\,r_\mathrm{E}$, is an example of a well-defined shock front where all criteria agree, and panel (f) shows an intermediate case where the three criteria disagree somewhat and the shock transition seems to extend radially over a distance of several hundred kilometers. An animation depicting time evolution of Figure 2 is available as Supplementary Video A.

We now describe how we evaluate the non-locality of the quasi-parallel bow shock in Vlasiator simulations. At one degree nose angle intervals, we draw a profile across the shock in the bow-normal direction, and measure where along the profile each of our three shock criteria (plasma density $n_\mathrm{p} = 2n_\mathrm{p,sw}$, solar wind core heating $T_\mathrm{core} = 4T_\mathrm{sw}$, and magnetosonic Mach number $M_\mathrm{ms} = 1$) indicate the local position of the shock is. Then, for each profile, we calculate the distance between the positions of bow shock parametrization closest and farthest from the Earth. This distance estimates the extent of the shock transition region, i.e., the non-locality of the shock. In Figure 3, panels (a) and (b), we plot stacked profiles displaying the temporal evolution of shock non-locality for simulations S1 and S2, respectively. Regions of enhanced shock non-locality appear to move along the shock front away from the nose region (indicated with a dashed line), as shown by the diagonal

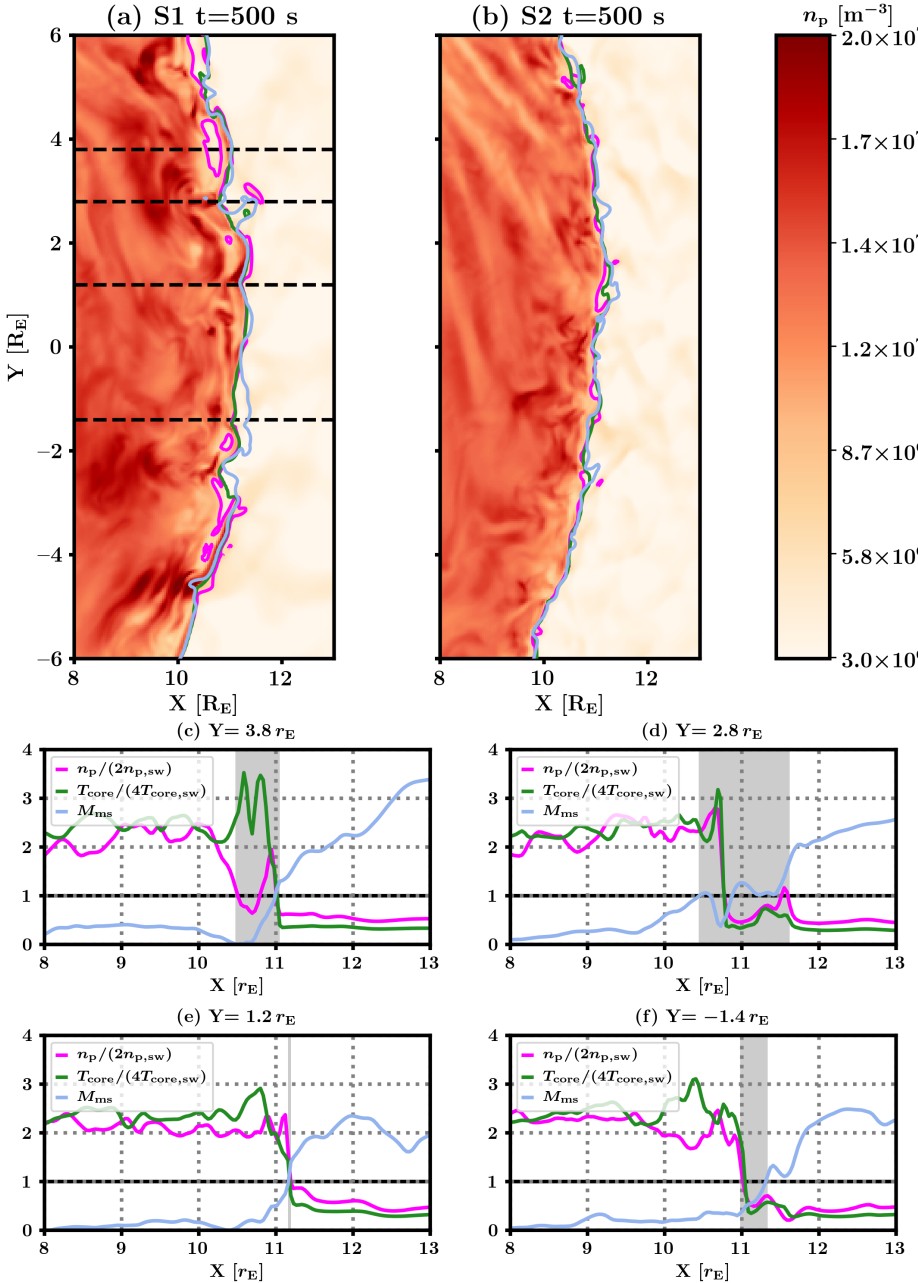

**Figure 2.** Proton number density overlaid with bow shock positions according to criteria for plasma density (fuchsia, $n_p = 2n_{p,sw}$), solar wind core heating (green, $T_{core} = 4T_{sw}$), and magnetosonic Mach number (pale blue, $M_{ms} = 1$). Panel (a) is for S1 ($B_{sw} = 5\,nT$), panel (b) for S2 ($B_{sw} = 10\,nT$), both at $t = 500\,s$. Panels (c–f) show line profiles of the three bow shock criteria along the dashed black lines shown in panel (a), corresponding with differing amounts of shock non-locality.

ridges. S1 shows significantly larger and clearer non-locality structures than S2. Still, there exists a qualitative similarity to the structures seen for both simulations. We note that the motion of structures away from the nose might be due to either deflected plasma flow carrying structures along the front, or due to foreshock wave fronts convecting in and interacting with a curved bow shock at increasing nose angle positions. This interaction is postponed to a further study. In panels (c) and (d), we show logarithmic histograms of accumulated shock non-locality measurements, showing that a well-defined shock is the most

common occurrence, and enhanced values of non-locality are increasingly rare. This also confirms that S2 has, on average, lower measurements of shock non-locality than S1 does.

Quantifying the non-locality of the quasi-parallel bow shock using spacecraft data will be more challenging than for simulations. Simulations allow us to directly measure spatial scales, whereas spacecraft motion in relation to quasi-parallel reformation is slow, and thus, use of a constellation of spacecraft and multipoint techniques are usually needed in order to infer spatial

scales. An estimate of spatial scales and non-locality can be achieved by evaluating the solar wind flow velocity, and multiplying that with the time difference between the first and last of the three presented metrics agreeing on being in the downstream. We note that this isn't a perfect measure, as showcased by Figure 2c where all three values are in agreement at $X = 11\,r_{\mathrm{E}}$ despite the density falling again drastically around $X \approx 10.6\,r_{\mathrm{E}}$. Constellation spacecraft measurements can however be used to verify the propagation direction of these structures, so a rarefaction within the sheath could be distinguished from a bow

shock moving inwards and outwards, increasing our understanding of shock reformation dynamics and the extent of the shock transition region.

## 4 Test-particle simulations

The Vlasiator model tracks the evolution of distribution functions as volume averages on a Cartesian mesh. Thus, particle trajectories are not a direct output of the code, and tracing particle histories requires the use of a post-processing tracer. In

order to evaluate injection probabilities, particles need to be tracked as they meet the bow shock and interact with it, ultimately either returning to the upstream or being transmitted to the downstream. Thus, we chose to use a test-particle method to track the motion of single protons within the evolving, locally interpolated electric and magnetic fields output from the Vlasiator simulation. The particle propagation uses a Boris-push algorithm (Boris, 1970) with a conservative time step of $\Delta t = 0.005\,\mathrm{s}$. This time step is not limited by particle gyrotimes, but rather, ensures that particles up to $10^5\,\mathrm{eV}$ travel less than $1/10^{\mathrm{th}}$ of

a simulation cell per time step. $\mathbf{E}$ and $\mathbf{B}$ field values for each particle step are acquired from the Vlasiator output files using linear interpolation in both time and space. Thus, the test-particles act as tracers for an infinitesimal element of the distribution function.

Our goal is to use test-particle simulations to investigate proton injection at the quasi-parallel bow shock. For this purpose, we initialize our particles from the thermal solar wind core population, evenly distributed along a smooth curve a short distance

in front of the bow shock. We follow the particles as they approach the shock region and interact with it. If a particle reaches again a boundary well in front of the shock, it is considered injected, and if it passes far into the downstream, it is considered transmitted. Once a particle has been flagged as injected or transmitted, it is no longer propagated. A significant portion of

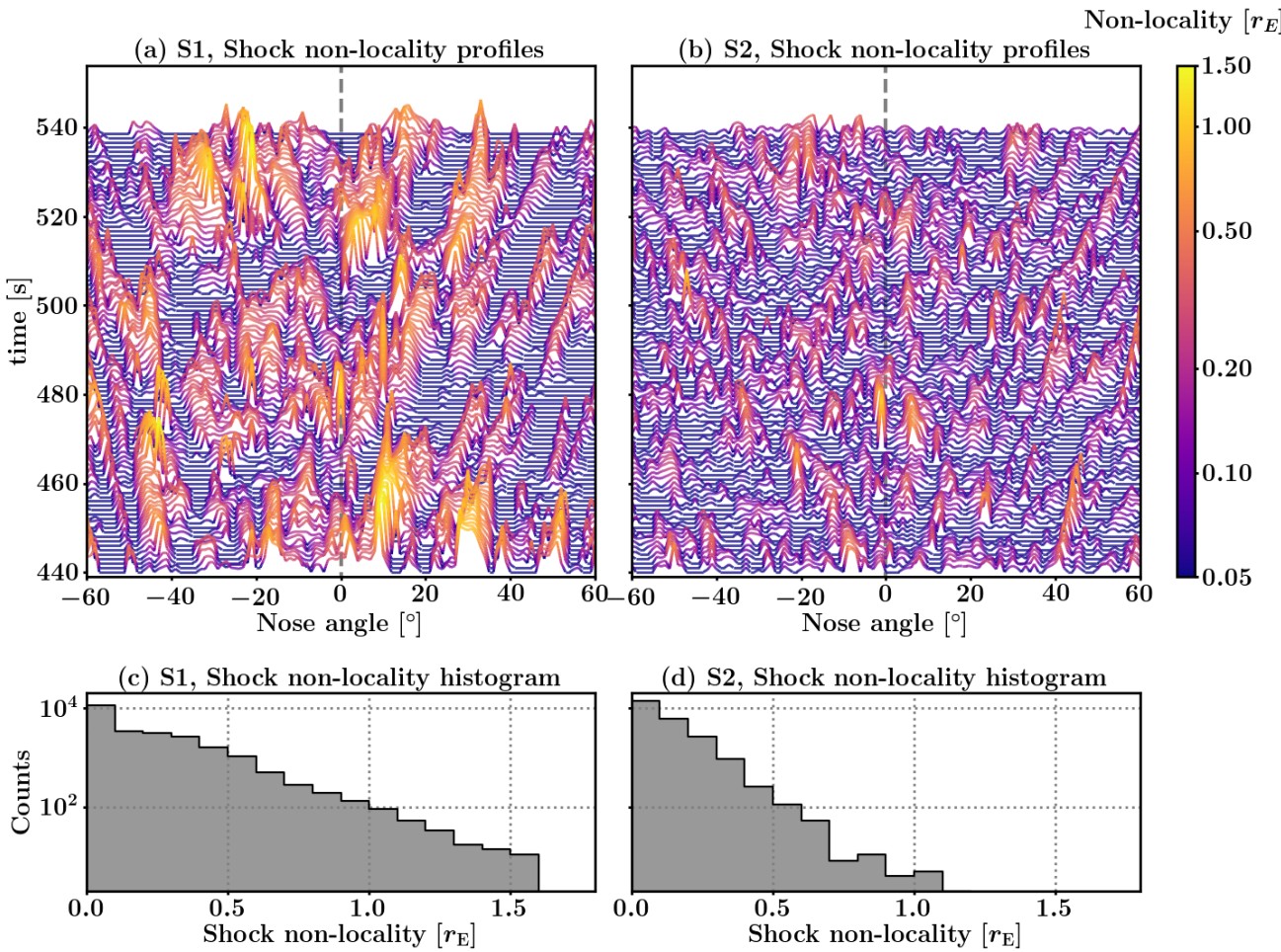

**Figure 3.** Panels (a) and (b): profiles of measured shock non-locality as a function of nose angle for simulations S1 and S2, respectively. The $y$-axis indicates simulation time, used as the base level corresponding with a well-defined shock with a non-locality measure of zero. Regions of enhanced non-locality are shown as colored peaks of the curve, as presented in the color bar. A dashed vertical line indicates nose angle $0°$. Both panels show chains of enhanced non-locality regions which move away from the nose region and decrease in intensity as they approach the flanks. Under each stacked profile plot we show a histogram depicting the occurrence rate of different non-locality values, with panel (c) depicting S1 and panel (d) depicting S2.

test-particles spend so much time within the shock structure that they are not flagged as either injected or transmitted at the end of the run, and their fate remains inconclusive.

The particle initialization curve is placed $0.9\,r_\mathrm{E}$ outward of the parabolic bow shock fit, extending between nose angles $\pm 40°$. This is visible in panel (a) of Figure 4 as the location of the first test-particles. An injection flagging boundary is placed $0.1\,r_\mathrm{E}$ beyond the injection curve, and a transmission flagging boundary is placed $1.5\,r_\mathrm{E}$ inward of the parabolic bow shock fit. These values were chosen so that the majority of changes to local quasi-parallel bow shock structure due to reformation fall within this region. We specifically note that the solar wind core heating criterion triggers always within this region.

Each test-particle run consists of $N = 10^5$ protons, initially isotropic in the frame co-moving with the inflow plasma, which results in a mean simulation frame energy of $1.9\,\mathrm{keV}$. For each test run, particle velocities were chosen as monoenergetic (10, 20, 50, 100, 200, or 500 eV) in the inflow plasma frame and randomly distributed in direction. Additionally a Maxwellian test run was performed, with particles picked randomly from a Maxwellian 0.5 MK distribution centered in the inflow plasma frame. For this distribution, the mean energy is 65 eV and the most probable energy is 43 eV. Particles were placed into the
simulation as groups of 25000 particles every 0.5 s for 10 seconds, starting at $t_0 = 438\,\mathrm{s}$. Particle propagation was halted at time $t_f = 538\,\mathrm{s}$.

## 5   Test-particle results

In Figure 4, we display snapshots of test-particle propagation for simulation S1 and an initially maxwellian distribution of 0.5 MK in the solar wind frame. The grayscale region shows a logarithmic test-particle density, with black indicating single
particles and white indicating over 100 particles per cell. We display contours parametrizing the shock position on top, and also plot two black parabolas which act as the injection and transmission flagging boundaries. Animations depicting the evolution of test-particle populations for all initialization parameters and simulations S1 and S2 are available in Supplementary Videos B and C, respectively.

    The panels in Figure 4 show how solar wind protons start as an even curve (a), are launched into the simulation over 10
seconds, after which the first ones have already accumulated as white regions at the shock front (b). We note how the steepened structure at $Y \approx 2\,r_\mathrm{E}$ in panel (b) causes an accumulation of test-particles at its $-Y$ edge, and that the regions of plasma depletion (fuchsia contour at, e.g., $Y \approx 6\,r_\mathrm{E}$, $Y \approx 2\,r_\mathrm{E}$, and $Y \approx -3\,r_\mathrm{E}$) remain void of test-particles at this time. By the time of panel (c), all test-particles have reached the shock transition region, the white regions of test-particle accumulation follow shock ripples, and many of the previously void regions have been filled with test-particles. In panel (d) we see regions of
efficient reflection causing particles to be returned to the upstream direction, but several regions also allow particles to move past the shock front, reminiscent of magnetosheath jets (Němeček et al., 1998; Hietala et al., 2009; Palmroth et al., 2018). By the time of panel (e), particles have spread to most of the magnetosheath all the way to the transmission boundary. Panel (f) displays how both transmission and injection can be slow processes, with 20–40% of particles still within the simulation after $90 - 100\,\mathrm{s}$ of test-particle propagation, both in the upstream and in the downstream of the shock. For these particles, their

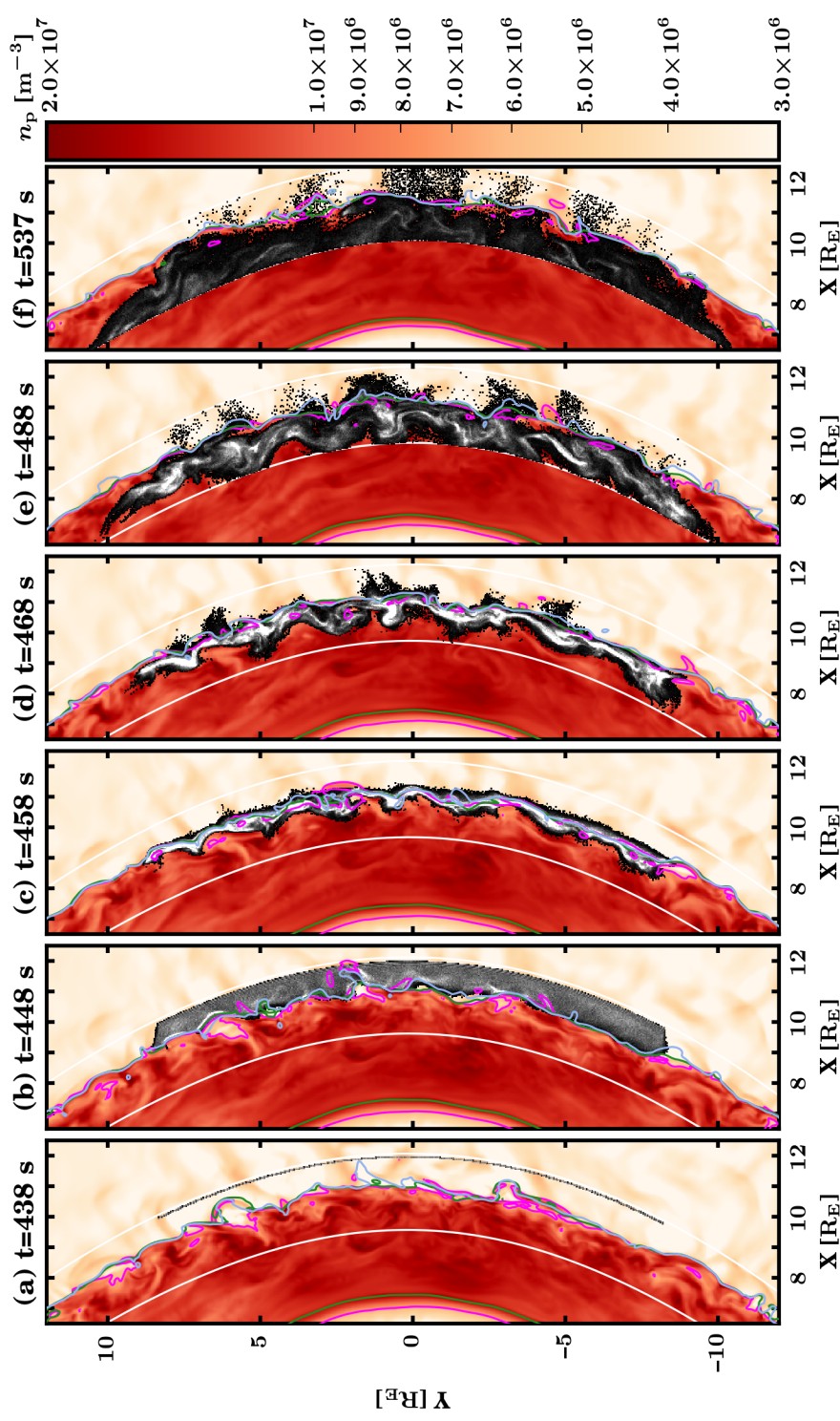

**Figure 4.** Test-particle propagation for simulation S1, maxwellian 0.5 MK initialization at 6 different times. Vlasiator simulation proton number density is overlaid with the logarithmic density of test-particles in greyscale, with white indicating over 100 particles in a cell. Two black parabolas are the transmission boundary (left) and the injection boundary (right). Three contours indicate estimates of the local shock position: plasma compression (fuchsia, $n_p > 2n_{p,sw}$), solar wind core heating (green, $T_{core} > 4T_{sw}$), and the magnetosonic Mach number (pale blue, $M_{ms} < 1$)

ultimate fate of being injected or transmitted could not be evaluated from these simulations. Judging from panel (f) of Figure 4, a portion of these particles would likely be injected.

Evaluation of test-particle interactions with the shock structure as seen in Figure 4 did not provide a quantitative answer as to where within the shock transition region particles truly feel the impact of the shock. As a particle injected into the upstream necessarily will experience energization in the solar wind frame, we tracked the solar wind frame energies of transmitted and injected particles and measured the regions where particles gained or lost the most energy. In Figure 5 we plot 2D-histograms of mean particle energy rate of change $\langle \Delta E / \Delta t \rangle$, which was calculated by measuring particle energy changes over 0.5 s intervals and finding the average by normalizing the result with the amount of test-particles measured at each position in parameter space. As energy gains and losses can be significant near strong electric fields (up to 1 keV per measurement interval), we use the energy preceding each interval as the y-coordinate. This emphasizes energy losses at high energies and energy gains at low energies. The black contours depict logarithmic counts of measurements, starting from a single particle with the thin dotted line. The count contours in panels c) and g) do not have a strong peak at the 10 eV initialization energy because those particles are advected efficiently towards the downstream whereas slightly energized particles can gyrate over a larger distance and enhance the relative counts around 100 eV.

We note that the energization colormap is a symmetric logarithmic plot, with a small linear region between $\pm 10\,\mathrm{eV\,s^{-1}}$. The presented initialization energies of 10 and 100 eV correspond to $44\,\mathrm{km\,s^{-1}}$ and $138\,\mathrm{km\,s^{-1}}$ plasma frame velocities, respectively. We show energization plots for only those particles which were registered as transmitted or injected by the end of the test-particle simulation. A grey band indicates the simulation frame solar wind ram energy, which is the minimum energy required for a particle to travel sunwards, and thus the minimum energy for injection (1.9 keV for the solar wind speed $600\,\mathrm{km\,s^{-1}}$). In the first two rows of Figure 5, the x-axis shows the distance $\Delta x$ from the closest instantaneous position where the solar wind core heating shock criterion is met. The last two rows plot the instantaneous shock non-locality for the measurement, extracted from the nose angle bins calculated in Section 3.2.

The top half of Figure 5 clearly shows how particles start at the bottom right corner of each panel at initialization energies and upstream of the shock, and how they on average gain energy as they approach the shock. In the downstream, energization of injected particles is very efficient up to about 10 keV and takes place over an extended distance instead of being constrained to a thin shock front at $\Delta x(T_{\mathrm{core}}) \approx 0$. Injected particles continue to gain energy in the whole downstream region, but begin to lose energy once back in the upstream. It is noteworthy that particles which end up injected can penetrate up to almost $1.5\,r_{\mathrm{E}}$ into the downstream before returning upstream, but that those particles are a minority, and at high energies and thus large gyroradii. These particles could perhaps be considered to be experiencing thermal leakage. Conversely, it is also possible that at least some of these measurements are made at times when the $T_{\mathrm{core}}$ estimate of the shock position has extended further upstream, making these particles appear to be further downstream than they actually are. The black contours depicting measurement counts show enhancement close to $\Delta x = 0$ and $E \gtrsim 1.9\,\mathrm{keV}$, consistent with those particles dwelling and being energized at the shock front. That area is also where injected particles may have their lowest simulation frame energies which would facilitate lingering at the shock front. Evaluating the particle count contours, we see that injected particles gain energy in the

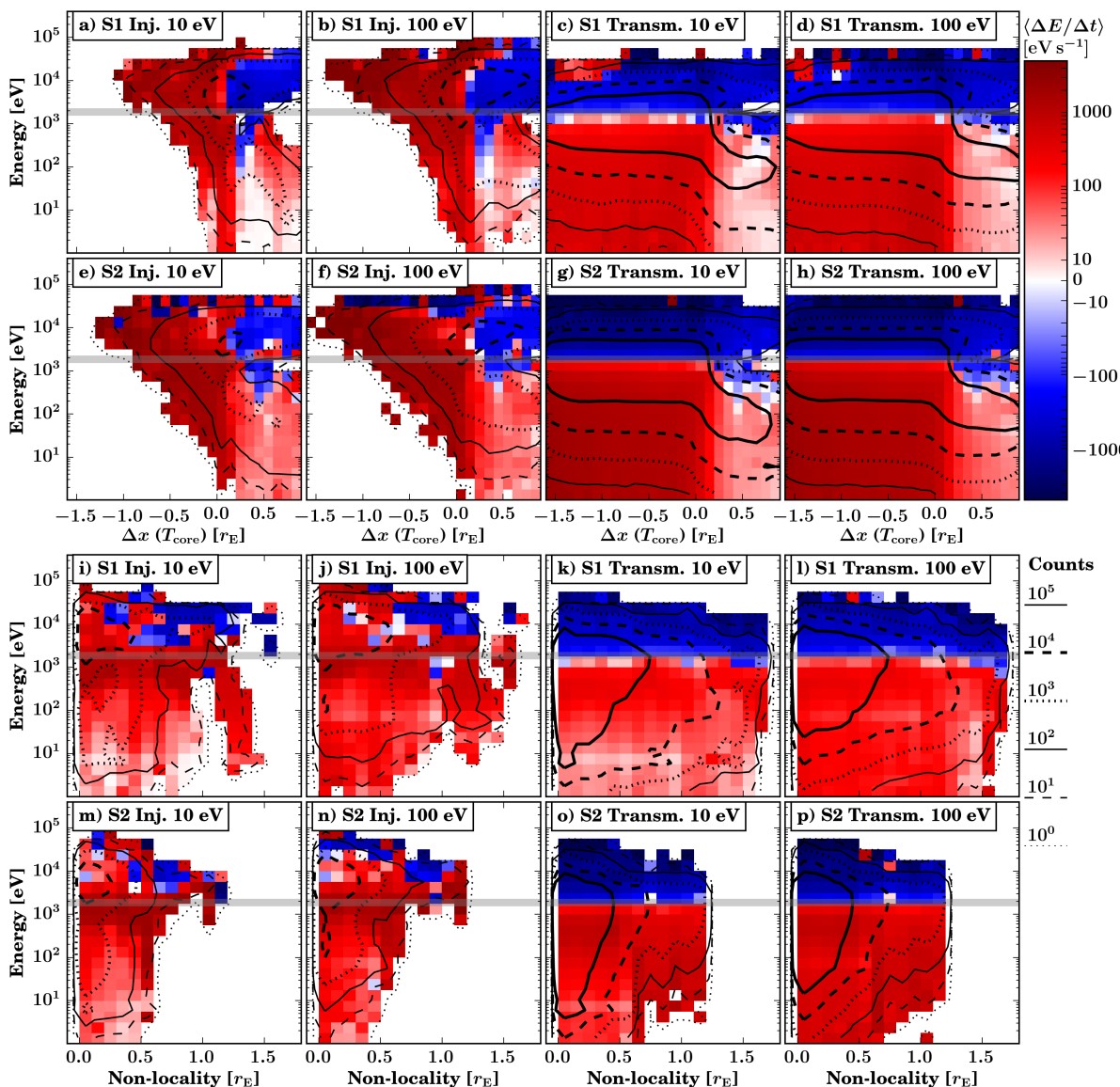

**Figure 5.** Mean energization experienced by test-particles over their shock interaction. Particle energy changes over 0.5 s intervals are integrated and averaged, recording them always at the pre-interval energy. Energization tracking is performed separately for injected (columns 1 and 2) and transmitted (columns 3 and 4) particles. The top two rows track energization as a function of current particle solar wind frame energy and $\Delta x$ from the closest position where the solar wind core heating shock criterion is met, and the bottom two rows as a function of current particle solar wind frame energy and shock non-locality. Rows 1 and 3 are from Simulation S1, rows 2 and 4 are from S2. Black logarithmic contours indicate the counts of measurements used for evaluating mean energization. A grey band indicates the minimum energy required for propagating upstream against the solar wind flow (1.9 keV for $600\,\mathrm{km\,s^{-1}}$). It is important to note that large values of shock non-locality can indicate signals of shock structure downstream as well as upstream of the parabolic shock fit position.

upstream as they approach the shock, but are not energized above the solar wind ram energy. The final required energization takes place in the downstream over a distance of up to $1.5\,r_{\mathrm{E}}$.

The behaviour of transmitted particles seen in Figure 5 is slightly different. They also start at the bottom right corner, at low energies and upstream of the shock, and experience energization already as they approach the shock. Throughout the downstream, these particles have a wide spread in energy and the dominant mechanism is to cool particles in the downstream rest frame, energizing (solar wind frame) low-energy particles and decreasing the energy of high-energy outliers. This is clearly visible as the split into blue (top) and red (bottom) halves of the panels. It should be noted that a small number of particles in the transmitted particles group are actually able to enter the upstream after exceeding the solar wind kinetic energy of 1.9 keV, but the efficient deceleration there returns them to the downstream and, ultimately, the transmission boundary. Both transmitted and injected particles are able to reach energies of up to $\sim$50 keV.

The two bottom rows of Figure 5 evaluate mean energization of test-particles as a function of energy and shock non-locality. Particle count contours show that the majority of measurements are made at regions where the shock is well defined, i.e., the non-locality measure is low. However, comparing these counts with the statistics of Figure 3 shows that there is little to no preference for particles spending time in regions of high or low shock non-locality. Although panels i, j, m, and n do not exhibit drastic energization preference for any single non-locality value, there are a number of conclusions to draw from them. At low energies ($E \lesssim 1\,\mathrm{keV}$), S1 shows an energization feature at non-locality values of $\sim$1.2 $r_{\mathrm{E}}$, whereas S2 indicates more efficient energization at non-localities at around $0.5\,r_{\mathrm{E}}$. This would indicate a connection with the inherent size of foreshock structures in the two runs, respectively (Turc et al., 2018). The majority of energization of injected particles happens once particles have reached energies of $E \gtrsim 1.9\,\mathrm{keV}$, allowing them to dwell in the vicinity of the shock and sample a wider range of non-locality values. Finally, at very high energies $E \gtrsim 10\,\mathrm{keV}$, a preference can be detected for energization at small values of non-locality and deceleration at large values of nonlocality, as indicated by the predominantly red and blue regions, respectively. For transmitted particles, there appears to be no clear indication of preferential energization parameter regions but we again detect that particles energized to 1.9 keV can sample a wider range of non-locality values.

Finally, we calculate injection probabilities $n_{\mathrm{inj}}/(n_{\mathrm{inj}} + n_{\mathrm{tra}})$ for test-particles in runs S1 and S2 as functions of a selection of parameters (detailed below) describing the first detected particle-shock-interaction. For each test-particle, we evaluate these properties at the first moment the particle reaches a point in the simulation space that fulfills the solar wind core heating ($T_{\mathrm{core}} > 4T_{\mathrm{sw}}$) criterion for the shock. Due to the non-locality of the quasi-parallel shock front, estimating when the particle-shock interaction is most significant is challenging, but we selected the one of our three methods which we visually estimated to be most meaningful (see also panels c–f of Figure 2).

In Figure 6, we plot the estimated injection probabilities for test-particle runs using S1 and S2, using the previously described test-particle data sets with six different solar wind frame initialization energies and a Maxwellian initialization. The first three rows use properties of particles in the simulation frame, namely the pitch-cosine $\mu = \cos(\alpha)$ (where $\alpha$ is the angle between the particle velocity and the local magnetic field direction), the incidence angle (the angle between the particle direction of travel and the opposite of the bow-normal direction $\angle(\mathbf{v}, -\hat{\mathbf{n}})$), and the shock-frame kinetic energy $E$. The last two rows of Figure 6 use shock properties, namely the local bow-normal angle $\theta_{Bn'}$ and the locally measured shock non-locality. Again,

these values were measured at the moment the particle first encountered the shock, according to the solar wind core heating criterion. Error bars are provided by the Agresti-Coull method with a 95% confidence interval.

The first row of Figure 6 indicates that if the particle encounters the shock with negative pitch-cosine, it is likely to be injected. In our simulation set-up, most particles travel roughly in the $-v_x$ direction, and with the IMF pointing roughly anti-sunward, most particles have pitch-cosines close to 1. Significant deviation from this suggests local magnetic field directions which have changed significantly due to foreshock wave effects. Our results indicate that these magnetic field deflections can enhance injection probabilities.

According to the second row, if the particle has a large incidence angle (the bow-normal velocity component is positive or small compared to the bow-perpendicular velocity component), injection is again likely. Incidence angles above $90°$ in fact suggest the particle was travelling away from the bow shock when it first met a shock structure. This could perhaps happen due to the particle gyrating along a deflected magnetic field line with a pitch-angle close to zero, so that its perpendicular velocity causes it to encounter a shock peninsula such as the one seen at $Y = 2.8\,r_E$ in Figure 2 from behind. We note that these plots show on average larger injection probabilities for higher plasma frame particle initialisation energies. This is as expected, as higher plasma frame initialization energies enable greater maximum energies when transforming into the spacecraft or simulation frame.

In the third row, we plot injection probabilities as a function of simulation frame energy, which corresponds very well with shock-frame energy due to the shock being mostly stationary on a global scale. This panel shows clearly how particles with greater initialization energies in the solar wind frame have a much larger spread in energy in the shock frame. Both very small and very large energies in the shock frame can lead to efficient injection. Small energies result in the particle spending much time at the shock, possibly then being accelerated in the shock frame with an upstream-directed velocity. Very large energies on the other hand mean that the particle does not need to be energized, it is enough to bend its path to the upstream in order to inject it. What we also see is that particles with a higher solar wind frame initialization energy tend to have a greater chance of being injected at a given shock-frame energy. These particles have a larger velocity component tangential to the shock, which suggests that being able to perform gyromotion in the fields at the shock is important for the injection and energization process.

The fourth row shows injection probability as a function of the local bow-normal angle $\theta_{Bn'}$. For S1, we see a small bump for low initialization energies at $\sim 70°$, and a significant increase at all energies at $\sim 85°$. Considering bow-normal angles above $90°$ may seem odd, but these regions are where foreshock fluctuations and shock effects have caused the local magnetic field to twist back on itself. For simulation S2, with a lower Mach number, these situations are not detected.

The fifth row indicates injection probability as a function of the shock non-locality measure at the moment the particle first encounters the plasma with $T > 4T_{\text{sw,core}}$. Both simulations S1 and S2 show a peak in injection probability at a non-locality value of $\sim 0.4\,r_E$, with even the lowest initialisation energies having a $\sim 10\%$ probability in S1. For simulation S1, there is a decline in injection probability as the non-locality value increases beyond $\sim 0.8\,r_E$, with an additional peak of injection at energies $> 100\,\text{eV}$ at $1.5\,r_E$. These peak positions are in rough agreement with the results of Figure 5, except for the $\sim 0.4\,r_E$ peak for S1. As that signal is very strong at all initialization energies, it it is most likely the result of a singular transient effect

causing a strongly deflected magnetic field. We note that it is not related to the incident particle gyroradius as it has a theoretical maximum value (for $v = v_\perp = 600\,\mathrm{km\,s^{-1}}$) of $0.2\,r_\mathrm{E}$ for S1.

As a final step, in Table 1 we display the overall calculated injection probabilities $N_\mathrm{inj}/(N_\mathrm{inj} + N_\mathrm{tra})$ per test-particle run for six test-particle initialization energies and a Maxwellian initialization. Due to the limited time period of test-particle propagation, at the end of the run a portion of particles were still within the shock transition zone. This is indicated by the completion ratio $(N_\mathrm{inj} + N_\mathrm{tra})/N_\mathrm{init}$. We find that the completion ratio for S1 rises somewhat with increasing intialization energy, but is very stable for S2. In agreement with expectations, the injection rate increases monotonically with greater initialization energies. The injection rates for Maxwellian distributions are located between the values for 50 eV and 100 eV initializations. The mean energy for 0.5 MK is approximately 65 eV. As a point of comparison, we also extracted the Vlasiator simulation suprathermal particle densities at positions $0.5\,r_\mathrm{E}$ and $1.0\,r_\mathrm{E}$ upstream of the shock, averaged over nose angles between $\pm 45°$ and between simulation times $t_0 = 438\,\mathrm{s}$ and $t_f = 538\,\mathrm{s}$. To facilitate comparison of these Vlasiator suprathermal particle densities $\langle n_\mathrm{p,st} \rangle$ with test-particle injection probabilities, the values are given in units of solar wind density and included as the final two rows of Table 1. We note that although the suprathermal particle density derives from the injection probability, it measures both freshly injected protons and those protons which have spent longer in the upstream. The order of Vlasiator S1 and S2 upstream suprathermal particle densities as a function of Mach number is thus opposite to that of test-particle injection probabilities. This effect is likely not an artefact of the test-particle method, but rather results from energetic particles being trapped in the upstream, interacting with the ULF waves. Although S2 is less efficient at injection, the foreshock wave-particle trapping interactions can cause reflected particles to spend extended periods of time in the upstream before returning to the shock. Supplementary videos B and C visualize the different dynamics between simulations S1 and S2. The suprathermal particle dynamics of S1 and S2 were investigated in Turc et al. (2018), as shown in their Figure 2, panels b through d.

## 6   Discussion

We now discuss our results presented in sections 3 and 5, attempting to clarify questions related to the non-locality of the quasi-parallel bow shock and thermal particle injection at the Earth's quasi-parallel bow shock. We note that our approach has a number of differences compared with previous shock injection studies. We make no pre-selection that particles must encounter the shock with only a single big energization like, e.g., Sundberg et al. (2016) do. We track particle injection based on a spatial boundary, instead of requiring the ion to achieve a given energy. In our simulation the mean solar wind energy or the shock ram energy is $E_\mathrm{ram} = \frac{m_i}{2}(M_A v_A)^2 \approx 1.9\,\mathrm{keV}$, and a requirement of 5–10 times this energy for particle injection (such as required by Caprioli et al., 2015) is met by approximately 40%-50% of our injected particles. We additionally note that the complicated global shock geometry used in our study prevents use of simple injection measures such as a positive $v_x$ component (Sundberg et al., 2016). We note that in modeling the cross-shock potential we neglect the electron pressure gradient term. The majority of the potential difference at the shock is, however, included in the Lorentz and Hall terms (Eastwood et al., 2007; Yang et al., 2009).

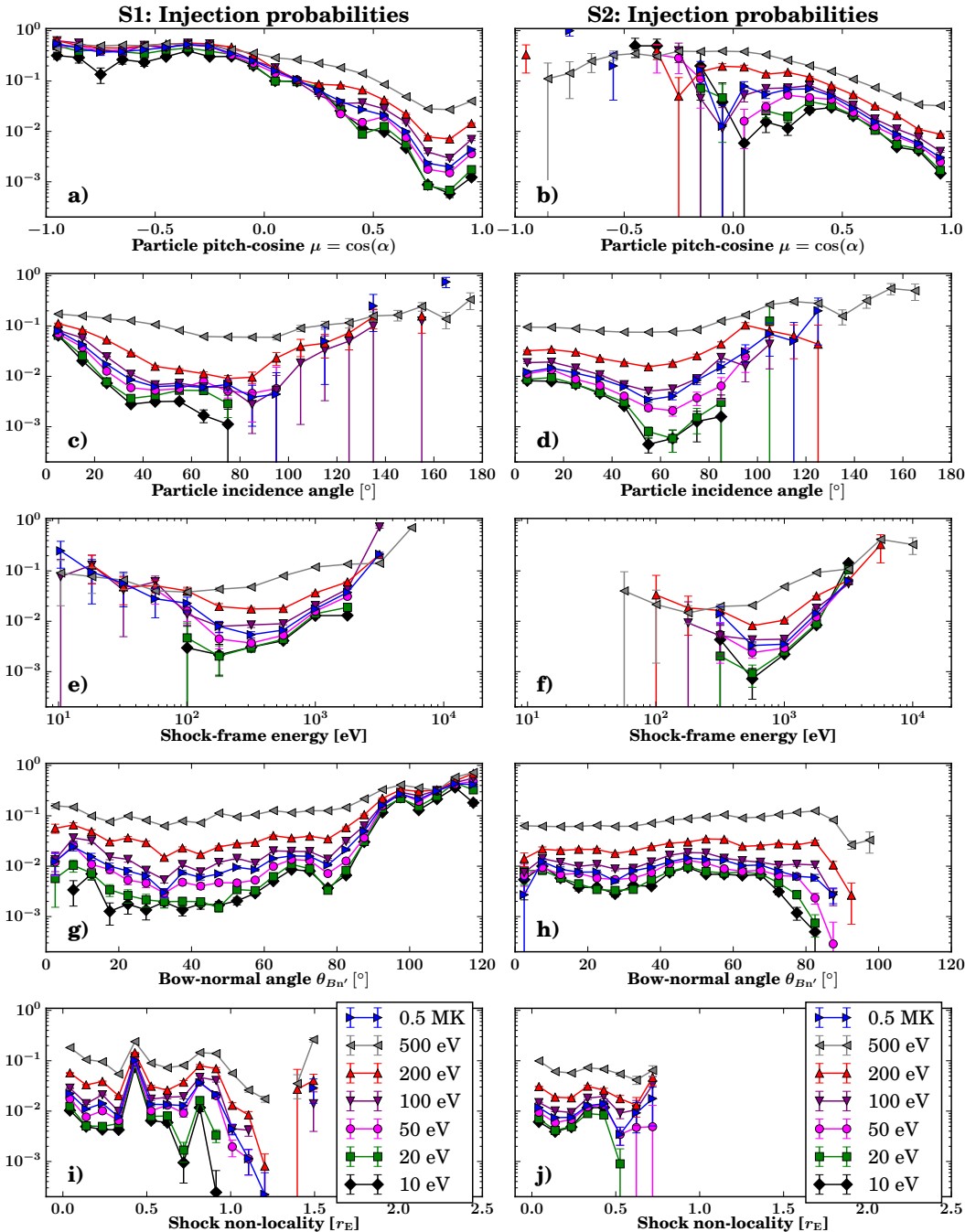

**Figure 6.** Test-particle injection probabilities for six different solar wind frame initialization energies and a $0.5\,\mathrm{MK}$ Maxwellian initialization and five different parameters. Left column: S1. Right column: S2. Rows 1 through 3 show properties of particles, namely the pitch-cosine $\mu = \cos(\alpha)$, the incidence angle, and the shock-frame energy. Rows 4 and 5 show shock properties, namely the local bow-normal angle $\theta_{Bn'}$ and the local shock non-locality. Displayed values were taken at the first encounter of each particle with the condition $T_{\mathrm{core}} > 4T_{\mathrm{core,sw}}$. Error bars are provided by the Agresti-Coull method with a 95% confidence interval.

**Table 1.** Test-particle proton statistics using simulations S1 ($M_A \approx 10$) and S2 ($M_A \approx 5$) with six different solar wind frame initialization energies $E_{\mathrm{init}}$ and also a Maxwellian initialization distribution with a temperature of 0.5 MK. Columns list the estimated injection probability $N_{\mathrm{inj}}/(N_{\mathrm{inj}} + N_{\mathrm{tra}})$ and the completion ratio $(N_{\mathrm{inj}} + N_{\mathrm{tra}})/N_{\mathrm{init}}$. Also shown is the ratio of injection probabilities for S2 and S1. The final two rows show suprathermal proton density measurements $\langle n_{\mathrm{p,st}} \rangle$ extracted from Vlasiator simulations S1 and S2, at positions $0.5\,r_{\mathrm{E}}$ and $1.0\,r_{\mathrm{E}}$ upstream of the mean bow shock position, averaged over nose angles between $\pm 45°$ and the test-particle run time extent.

| Test-particle $E_{\mathrm{init}}$ | S1 injection | S1 completion | S2 injection | S2 completion | S2/S1 injection ratio |
|---|---|---|---|---|---|
| 10 eV | 0.011 | 0.58 | 0.0058 | 0.79 | 0.53 |
| 20 eV | 0.013 | 0.59 | 0.0063 | 0.79 | 0.48 |
| 50 eV | 0.018 | 0.59 | 0.0086 | 0.79 | 0.48 |
| 100 eV | 0.027 | 0.60 | 0.013 | 0.78 | 0.48 |
| 200 eV | 0.047 | 0.62 | 0.027 | 0.77 | 0.75 |
| 500 eV | 0.13 | 0.67 | 0.085 | 0.77 | 0.65 |
| Maxwellian | 0.021 | 0.59 | 0.010 | 0.78 | 0.48 |
| Vlasiator suprathermals | $\dfrac{\langle n_{\mathrm{p,st}}(\mathrm{S1}) \rangle}{n_{\mathrm{p,sw}}}$ | | $\dfrac{\langle n_{\mathrm{p,st}}(\mathrm{S2}) \rangle}{n_{\mathrm{p,sw}}}$ | | $\dfrac{\langle n_{\mathrm{p,st}}(\mathrm{S2}) \rangle}{\langle n_{\mathrm{p,st}}(\mathrm{S1}) \rangle}$ |
| at $r_{\mathrm{shock}} + 0.5\,r_{\mathrm{E}}$ | 0.042 | | 0.061 | | 1.45 |
| at $r_{\mathrm{shock}} + 1.0\,r_{\mathrm{E}}$ | 0.027 | | 0.037 | | 1.37 |

Examination of Figure 2 shows that the spatial structure of bow shock non-locality depends on the magnitude of the upstream magnetic field, and thus, the spatial scale of foreshock structures. In Figure 3, it is evident that S1 shows clearer structures and stronger peaks of non-locality. The fine structure seen in S2 is as expected due to the increased magnetic field strength, which gives rise to smaller-scale structures in the foreshock and higher frequencies for the ULF waves (Turc et al., 2018), which in turn are expected to drive shock reformation. We suggest that spacecraft measurements of bow shock crossings could be evaluated using our definition of non-locality, inferring tendencies for the non-locality of the bow shock versus, e.g., IMF conditions and position nose angle. Although our method was defined as a function of radial distance, it should be applicable for spacecraft time series as well as suggested in section 3.2.

We also investigated the energization taking place during the first shock encounter of protons, before diffusive acceleration per se. We found that protons were weakly energized over a large distance as they approached the shock, that strong energization took place at the shock and over a distance of up to $1\,r_{\mathrm{E}}$ in the downstream, and that those protons which returned to the upstream experienced solar wind frame energy losses over the whole upstream region. Particles did, however, dwell for longer at the mean shock front position (panels a, b, e, and f of Figure 5). We found that the majority of injected particles did not penetrate far into the downstream, but a few did, and as they had achieved high energies, they might constitute injection through thermal leakage from the downstream. As we initialized our particles isotropic in the upstream plasma frame, we could see that particles which had simulation frame energies well below the solar wind energy were actually preferentially injected, similar to

the SLAMS reflection test-particle studies of Johlander et al. (2016) (see panels e and f of Figure 6). Protons with shock-frame particle energies close to the solar wind ram energy were more likely to be transmitted.

Interestingly, our result of energization taking place over a large area somewhat contradicts the results of, e.g., Guo and Giacalone (2013), who in simulations of a $M_A = 4$ shock saw initial energization very close to the shock (within $\sim 10 \, c/\omega_{ci}$ of the shock, or in our nomenclature, $\sim 0.2 \, r_E$). The difference may be caused by our integral energization tracking method differing from their method. The size of bow shock reformation in our simulation is (at $\sim 50 \, c/\omega_{pi}$) in agreement with the results of, e.g., Omidi et al. (2013) and Caprioli and Spitkovsky (2013).

We also evaluated particle energization as a function of shock non-locality. For the most part, energization rates appear to be equal at all non-locality values, although at low energies each simulation showed increased energization at a non-locality length scale which appears related to the spatial scale of foreshock structures. This result supports the theory of protons being efficiently energized between the existing shock and incoming shocklets/SLAMS or steepened ULF waves.

Statistical analysis of correlations between shock and particle properties and injection probability is presented in Figure 6. The most obvious result is that there are very few injected particles at large incidence angles, especially at lower initialization energies. For S1, there appears to be a connection between enhanced injection probability and incidence angles close to zero. A small incidence angle will likely correlate with greater-than-average simulation frame initialization energy, and higher energy is known to increase injection probability. We also reported on an increase in injection probability both with increasing solar wind frame energy and with shock frame energy diverging from the solar wind ram energy.

The fourth row of Figure 6 highlights the importance of magnetic field deflections upstream and at the shock for efficient particle injection. Simulation S1 with the higher Mach number and larger foreshock structures is much more efficient at forming strong deflections, resulting in bow-normal angles of above $80°$, whereas they are absent in S2. We emphasize that these measurements were performed within the globally quasi-parallel region of the bow shock, between nose angles $\sim \pm 40°$. We also note that in S1, there is an increase in injection at low initialization energies for bow-normal angles $\leq 15°$. This is likely the same effect as what Sundberg et al. (2016) described as injected ions encountering a locally quasi-perpendicular field downstream of the shock. This also warrants further investigation. Strong deformation of magnetic fields can also lead to other forms of energization such as localized reconnection found in the quasi-parallel shock transition region (Gingell et al., 2019). Resolving these effects appears to require higher resolution simulations.

The fifth row of Figure 6 evaluates the link between shock front non-locality and proton injection. S1 exhibits a peculiar peak in injection probability at $\sim 0.4 \, r_E$, which we presume to be due to a reformation-associated transient. S2 does not exhibit large non-locality values, but for S1, injection probability seems to fall past values $\sim 0.9 \, r_E$, with another peak at $\sim 1.5 \, r_E$. At low initialization energies, injection probabilities appear to fall off faster with increasing non-locality of the shock. Similar to Figure 5, slight enhancements in injection can be seen at non-locality values which appear related to the size of foreshock structures in the vicinity of the shock front. We propose that the lack of a strong link between injection and non-locality shows how shock injection at a curved reforming quasi-parallel shock is a complicated process, and local two-dimensional simulations showing clear-cut cyclical reformation and injection are capable of investigating only an idealized subset of reforming shock fronts.

We finally note that on time scales represented in our test-particle simulations, local structures of the quasi-parallel bow shock do have a significant effect on particle injection at all initialization energies. This is likely akin to what, e.g., Hao et al. (2017) and Sundberg et al. (2016) reported on, with rippled shapes of the shock front and advected magnetic fluctuations resulting in regions of localized injection. At the same time, our results suggest that energization of injected particles takes place over an extended region both at the shock and especially downstream of it. Thus, injection of ions at a quasi-parallel shock may happen via multiple different routes and phenomena.

The overall injection probabilities inferred from our test-particle studies agree with the strength of the shock (and the Alfvénic Mach number) indicating the overall injection probability of the shock. However, we note that the suprathermal particle density registered in the upstream of the shock did not agree with this result, indicating that the evolution of suprathermal particle populations throughout the foreshock is a complicated process and not a simple indicator of local shock reflectivity. One important effect to note is that of particle trapping between foreshock waves, as reported by Wu et al. (2015). We suggest that when performing studies of shock reflectivity using spacecraft measurements, extra care should be taken to differentiate freshly injected particles from an evolved foreshock population.

## 7 Conclusions

We have investigated the dynamics of the reforming quasi-parallel bow shock of the Earth in connection with the injection of thermal solar wind protons, using both hybrid-Vlasov and test-particle studies. Our noise-free hybrid-Vlasov simulations have allowed us to probe the reforming quasi-parallel bow shock dynamics in greater detail than previously possible, accounting for correct scale separation, the global dynamics of bow shock curvature, and for effects stemming from tenuous upstream particle distributions. Our results have shown that the energization and injection of solar wind ions within this region are not local effects taking place at a single shock location, but rather, are spread out over a larger shock transition region spanning up to $1.5\,r_{\mathrm{E}}$. We confirm enhanced particle injection with higher Alfvénic shock Mach number and plasma frame particle energy as expected. We also find that whenever the shock-associated magnetic field is deflected a great deal, particle injection is enhanced. A weak enhancement could also be seen in one of our simulations at very small bow-normal angles $\theta_{Bn'}$, so the interaction of magnetic field directions just upstream and downstream of the shock requires further study.

In our investigation, we defined a new metric for the bow shock, indicating the magnitude of non-locality of the shock front, associated with reformation. This metric was seen to correlate with the parameters of the foreshock and associated fluctuations, and also thus the shock Alfvénic Mach number. We showed how enhancements of non-locality travelled away from the shock nose and towards the flanks, indicating persistent interaction between the upstream ULF wave field and the shock front. We found that energization of cool solar wind frame particles was not dependent on a specific value of shock non-locality, which is in agreement with our finding of particle energization within the quasi-parallel bow shock region taking place over a large extent, not only at the shock front. At very high energies $E \gtrsim 10\,\mathrm{keV}$, some preference was seen for particle energization at small values of non-locality. Although the metric was defined as a spatial measurement, it can be applied to spacecraft

measurements and used to investigate the effect of shock reformation on energization of injected particles, particularly at high energies.

Our study concentrated on two bow shock simulations, so additional studies into the locality of injection and energization of solar wind particles using a more extensive simulation database are warranted.

We further note that the local density of suprathermal particles may be a poor indicator of injection efficiency of the shock due to large-scale dynamics of the foreshock region, such as particle trapping. This is an important factor when using either simulation results or spacecraft observations for estimating injection efficiencies at the bow shock.

*Code and data availability.* Vlasiator (http://www.physics.helsinki.fi/vlasiator/, Palmroth, 2020) is distributed under the GPL-2 open source license at https://github.com/fmihpc/vlasiator/ (Palmroth and the Vlasiator team, 2020). Vlasiator uses a data structure developed in-house (https://github.com/fmihpc/vlsv/, Sandroos, 2019), which is compatible with the VisIt visualization software (Childs et al., 2012) using a plugin available at the VLSV repository. The Analysator software (https://github.com/fmihpc/analysator/, Hannuksela and the Vlasiator team, 2020) was used to produce the presented figures. The run described here takes several terabytes of disk space and is kept in storage
maintained within the CSC – IT Center for Science. Data presented in this paper can be accessed by following the data policy on the Vlasiator web site.

*Video supplement.* The Supplementary Videos A, B, and C provide movie extensions of Figures 2 and 4, showcasing the evolution of the quasi-parallel shock front profiles and the associated non-locality (Video A) and the evolution, transmission, and injection of test-particle populations of various initialization parameters for simulations S1 (Video B) and S2 (Video C).

Movie A (Battarbee et al., 2020a). Movie extension of Figure 2. Animation of proton number density overlaid with bow shock positions according to criteria for plasma density (fuchsia, $n_\mathrm{p} = 2n_\mathrm{p,sw}$), solar wind core heating (green, $T_\mathrm{core} = 4T_\mathrm{sw}$), and magnetosonic Mach number (pale blue, $M_\mathrm{ms} = 1$). Panel (a) is for S1 ($B_\mathrm{sw} = 5\,\mathrm{nT}$), panel (b) for S2 ($B_\mathrm{sw} = 10\,\mathrm{nT}$), both at $t = 500\,\mathrm{s}$. Panels (c–f) show line profiles of the three bow shock criteria along the dashed black lines shown in panel (a), corresponding with differing amounts of shock non-locality.

Movie B (Battarbee et al., 2020b). Movie extension of Figure 4. Test-particle propagation for simulation S1 ($B_\mathrm{sw} = 5\,\mathrm{nT}$), with 6 different monoenergetic initialization as well as a Maxwellian 0.5 MK initialization. Vlasiator simulation proton number density is overlaid with the logarithmic density of test-particles in greyscale, with white indicating over 100 particles in a cell. Two black parabolas are the transmission boundary (left) and the injection boundary (right). Three contours indicate estimates of the local shock position: plasma compression (fuchsia, $n_\mathrm{p} > 2n_\mathrm{p,sw}$), solar wind core heating (green, $T_\mathrm{core} > 4T_\mathrm{sw}$), and the magnetosonic Mach number (pale blue, $M_\mathrm{ms} < 1$.

Movie C (Battarbee et al., 2020c). Movie extension of Figure 4. Test-particle propagation for simulation S2 ($B_\mathrm{sw} = 10\,\mathrm{nT}$), with 6 different monoenergetic initialization as well as a Maxwellian 0.5 MK initialization. Vlasiator simulation proton number density is overlaid with the logarithmic density of test-particles in greyscale, with white indicating over 100 particles in a cell. Two black parabolas are the transmission boundary (left) and the injection boundary (right). Three contours indicate estimates of the local shock position: plasma compression (fuchsia, $n_\mathrm{p} > 2n_\mathrm{p,sw}$), solar wind core heating (green, $T_\mathrm{core} > 4T_\mathrm{sw}$), and the magnetosonic Mach number (pale blue, $M_\mathrm{ms} < 1$.

*Author contributions.* MB carried out the analysis and wrote the manuscript. UG and YPK have made significant contributions to the simulation methods and analysis. All co-authors helped in the interpretation and vizualisation of the results, read the manuscript, and commented on it.

*Competing interests.* The authors declare that they have no conflict of interest.

*Acknowledgements.* We acknowledge the European Research Council for Starting grant 200141-QuESpace, with which Vlasiator (Palmroth, 2020) was developed, and Consolidator grant 682068-PRESTISSIMO awarded to further develop Vlasiator and use it for scientific investigations. The Finnish Centre of Excellence in Research of Sustainable Space, funded through the Academy of Finland grant number 312351, supports Vlasiator development and science as well. We also gratefully acknowledge the Academy of Finland (grant number 267144). The CSC – IT Center for Science in Finland is acknowledged for the Sisu supercomputer usage and Grand Challenge award leading to the results presented here. The work of L. Turc was supported by a Marie Sklodowska-Curie Individual Fellowship (#704681). We wish to thank Lynn B. Wilson III and Andreas Johlander for fruitful commentary on this topic.

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
