# Peer review of "Non-locality of the Earth's quasi-parallel bow shock: injection of thermal protons in a hybrid-Vlasov simulation"

_Annales Geophysicae, 2019_

## Author Comment (AC1) · 11 Sep 2019

Unfortunately, the axes and labels in Figure 6 of the discussion manuscript have fallen victim to some kind of encoding error and have become illegible. Please accept our apologies for this, and find the original Figure 6 attached to this message. Following revisions of the manuscript shall be double-checked to ensure the images are correctly embedded.

Best regards,
Markus Battarbee & co-authors

[Figure]

**Figure 6**: Test-particle injection probabilities for six different solar wind frame initialization energies and a $0.5\,\mathrm{MK}$ Maxwellian initialization and five different parameters. Left column: S1. Right column: S2. Rows 1 and 2 show properties of particles, namely the pitch-cosine $\mu = \cos(\alpha)$ and the incidence angle. Rows 3 through 5 show shock properties, namely the local bow-normal angle $\theta_{Bn'}$, the local shock porosity, and the impact position nose angle. Error bars are provided by the Agresti-Coull method with a 95% confidence interval.

[Figure]

**Fig. 1.** Figure 6 of manuscript

[Figure]

---

## Referee Comment (RC1) · Anonymous Referee #1 · 26 Sep 2019

The manuscript by Battarbee et al. has discussed the proton injection issue with results obtained by global hybrid-Vlasov and test-particle simulations. I think the quality of the paper is more than enough for publication. I have a few comments and suggestions that the author might want to address before the paper should be published.

The word "non-locality" is a little bit confusing. I think it is more or less similar to the thickness of the shock (although not necessarily the same). It might be better to add some explanation for this as it is not in the standard terminology.

According to the description of the simulation parameters, the spatial resolution (228 km) is larger than the ion inertial length (125 km). It may not be so bad for modeling

global phenomena, but one must be careful for doing accurate simulations of collision-less shocks. In particular, since the authors followed test particle trajectories on top of their simulation results to discuss the particle interaction with the shock, the resolution can be an issue. I guess that it is not easy to perform a convergence study for this particular application in a reasonable amount of computational resources. However, the authors may caution to the readers that there is potentially a numerical resolution issue. The disagreement between the Vlasiator and test-particle results in table 1 may also arise from the same reason.

It is no surprise to me that the non-locality is not an important factor to affect the injection as the ions have long interaction time with the shock and can travel for a long distance along the shock surface before being reflected or transmitted. The fate of the particles should be determined by the integral of electromagnetic fields as seen by them.
* * *

---

## Referee Comment (RC2) · Anonymous Referee #2 · 9 Dec 2019

The manuscript analyses Vlasov numerical simulations including test particle runs of the quasi-parallel Earth's bow shock and ion acceleration there. The concept of "shock non-locality" is introduced and it is shown that the non-locality has little direct effect on particle injection. Instead the injection takes place in a larger region surrounding the shock but at the same time local magnetic field distortions at the shock are important for the injection.

Abstract promises a novel method for spacecraft data analysis. It is not clear what is the novel method, why and how it should be applied and what would be the outcome. This needs to be clarified.

[Figure]

Simulation initialisation: - Why 5° tilt in the magnetic field? - Is 43eV solar wind plasma temperature motivated by the velocity resolution? - There should be a proper discussion why simulating the system with a simulation having the spatial resolution larger than the characteristic ion inertial and ion gyroradius scales is appropriate to address the problem of ion injection where most of the ion reflection can occur on ion kinetic scales. This points needs to be clearly addressed as it may affect the general conclusions of the paper.

It is not clear why authors have chosen to do the test particle approach if the Vlasov code is supposed to follow the full distribution function. This point needs to be clearly explained.

The concept of non-locality is introduced which does not include magnetic field. The motivation is that it provides poor results while at the same time paper mentions that magnetic field structures are very important for the injection. All this makes the motivation for the non-locality concept very unsatisfactory and it is not clear what authors mean motivating their selection by having poor vs good results. Magnetic field data is one of the primary datasets in the shock analysis and it is unclear why one would want to exclude it from the shock definition. In general, it is not clear why authors want to introduce a new concept.

In Figure 4 100eV case of test particles is shown. It should be motivated why this particular case is shown and not for example the case of Maxwellian distributed particles.

Figure 5 results and discussion are not fully consistent and should be significantly improved. For example, showing injected particle results (column 1 and 2) one makes conclusion that particles with energies below solar wind drift energy are loosing energy on average and particles above are gaining. This results is inconsistent with that the figures shows most of the injected particles have high energy. Such high energy particles if they start at solar wind energy and then during some part of the orbit have energies below the solar wind energy then on average at low energies the energy gain and loss

should be equal. If the statement made in the manuscript is true then why there are no low energy injected particles (while there are still a lot of low energy particles at r<0 and they all show negative energy gain.

Similarly, it is not clear how the current simulations results contradict the results from the Johlander et al. Firstly, it is not clear if SLAMS are observed in the current simulation and if they are do they have similar properties as in the observations? Secondly, when comparing with Johlander et al., it would be good to do the comparison in an adequate way, so that one understand how one should translate the results from the Vlasiator case to another cases such as Johlander et al. For comparison with those results one would need to look at solar wind ions that have different kinetic energy in the shock frame and see the differences in the injection rate. The authors should guide the reader where and how this can be seen.

Figure 6 requires several clarifications. The largest structuring of the injection probabilities is seen in the dependance on the impact position angle. Instead of trying to resolve the physics of the large injection rate variations authors suggest how to smooth these variations which suggests that authors themselves maybe do not trust the numbers. This needs to be clarified. Another unclear point is how shock non-locality is defined for a particle that starts at one position and gets injected at another position (in general valid for all particles). From which time and position are the given shock non-locality values. Similarly, it is not clear at which time instant is measured the bow-normal angle.

Minor things: L.8 fix the language of the sentence.

l.45 The work of Johlander et al. 2016 does not make the mentioned assumptions in the manuscript but shows that SLAMS can contribute to the injection.

l.59 It is a bit confusing in which reference frame particle gains energy in the definition of the energization. For example, a particle reflected from a shock can have lower energy in the shock reference frame than the solar wind particles (e.g. when reflected from SLAMS) but it would not be "part of the incident thermal distribution".

l.165 the division of the core distribution is unclearly described.

l.368 this should be illustrated and quantified, adding by the figure

l.411 What do you mean by "high-fidelity"?

Figure 1: please use slightly thinner lines, the structure of the pink line cannot be resolved in the figure due to the thickness of the line. '

Table 1: Why comparison is done with suprathermal densities? I assume that from Vlasiator one can estimate the flux of reflected particles and thus have a good estimate of the injection rate.

---

## Author Comment (AC2) · 20 Dec 2019

*The manuscript by Battarbee et al. has discussed the proton injection issue with results obtained by global hybrid-Vlasov and test-particle simulations. I think the quality of the paper is more than enough for publication. I have a few comments and suggestions that the author might want to address before the paper should be published.*

We thank the referee for the review and improvement suggestions.

*The word "non-locality" is a little bit confusing. I think it is more or less similar*

*to the thickness of the shock (although not necessarily the same). It might be better to add some explanation for this as it is not in the standard terminology.*

We agree that coming up with terminology for a new concept is challenging, and acknowledge how, in some respects, non-locality is similar to a shock thickness. We propose that a thickness is really only valid when the shock has a well-defined upstream and downstream and a clear transition between them, e.g. in the context of the quasi-perpendicular shock when the shock profile is clear and unambiguous (well localized). In the quasi-parallel region there are challenges associated with finding the shock profile, in particular as the shock reforms, as shown in Figure 2. We will add a comparison to shock thickness to the terminology subsection in the introduction.

*According to the description of the simulation parameters, the spatial resolution (228km) is larger than the ion inertial length (125 km). It may not be so bad for modelling global phenomena, but one must be careful for doing accurate simulations of collision-less shocks. In particular, since the authors followed test particle trajectories on top of their simulation results to discuss the particle interaction with the shock, the resolution can be an issue. I guess that it is not easy to perform a convergence study for this particular application in a reasonable amount of computational resources. However, the authors may caution to the readers that there is potentially a numerical resolution issue.*

We appreciate the reviewer's concern regarding the ion inertial length. A convergence test is indeed unrealistically expensive to perform. We do, however, intend to investigate this issue in the future.

We would also like to note that there exists a trade-off in simulations which focus on the small scales. For example, the mesoscale reformation features shown in our Figure 2 panel a) can have spatial extents of up to 2 RE or 100 di. These arise from the interaction of the curved bow shock with incident ULF wave fronts. With a given set of simulation resources, one needs to either run a local simulation, perform system rescaling (e.g. Tóth et al 2017) which will negatively impact the global dynamics, or have a spatial resolution which does not resolve effects at or below ion inertial length scales. Our approach aims to investigate effects arising from the global scale. We intend to elaborate this approach and our motivation in our manuscript.

We also note that the qualitative bow shock effects and reformation seen in these simulations are in agreement with other Vlasiator simulations (see the web site, ) where the 30 degree IMF simulation cell size was set to the ion inertial length. We will investigate this run in the future, but wanted to utilize the quasi-parallel IMF for this initial study.

***The disagreement between the Vlasiator and test-particle results in table 1 may also arise from the same reason.***

Since the test-particles and the Vlasiator distribution functions both are acted upon by fields of identical spatial resolution, we did not consider this a likely cause for the discrepancy. Test-particle fields are interpolated on the subgrid level in a linear fashion whereas the Vlasov distributions use volumetric-reconstructed fields.

The wave fields in the two runs are very different (Turc et al. 2018), resulting in differing trapping dynamics. We plan to elaborate this point of discussion accordingly.

***It is no surprise to me that the non-locality is not an important factor to affect the injection as the ions have long interaction time with the shock and can travel for a long distance along the shock surface before being reflected or transmitted. The fate of the particles should be determined by the integral of electromagnetic fields as seen by them.***

We agree that the electromagnetic fields are indeed the key to evaluating particle injection. As particle injection time scales are indeed significant, and close to reformation time scales, we felt it important to investigate a possible connection. There does not yet exist consensus in the field for injection and shock physics, but hopefully future studies will find convergence in our understanding of the quasi-parallel plasma shock.

---

## Author Comment (AC3) · 20 Dec 2019

We wish to thank the referee for the helpful review.

*The manuscript analyses Vlasov numerical simulations including test particle runs of the quasi-parallel Earth's bow shock and ion acceleration there. The concept of "shock non-locality" is introduced and it is shown that the non-locality has little direct effect on particle injection. Instead the injection takes place in a larger region surrounding the shock but at the same time local magnetic field distortions at the shock are important for the injection.*

[Figure]

*Abstract promises a novel method for spacecraft data analysis. It is not clear what is the novel method, why and how it should be applied and what would be the outcome. This needs to be clarified.*

Thank you, we will clarify that the proposed new analysis is calculating non-locality from a combination of three plasma measurements.

*Simulation initialisation:*
*- Why 5° tilt in the magnetic field?*

In this study, we chose to focus on the quasi-parallel bow shock. The 5 degree run was chosen as it provided a large region where global curvature effects were captured within the quasi-parallel bow shock.

*- Is 43eV solar wind plasma temperature motivated by the velocity resolution?*

This is correct. In order to ensure the shock dynamics are properly modelled, the incoming solar wind distribution must be adequately resolved by the velocity grid. This is verified by ensuring there is no numerical heating as the distribution propagates from the inflow boundary to the shock.

*- There should be a proper discussion why simulating the system with a simulation having the spatial resolution larger than the characteristic ion inertial and ion gyroradius scales is appropriate to address the problem of ion injection where most of the ion reflection can occur on ion kinetic scales. This points needs to be clearly addressed as it may affect the general conclusions of the paper.*

We thank the referee for the feedback. We intend to improve this discussion in the paper with the following reasoning. Our simulations choose to emphasize the global dynamics due to, e.g. curved bow shock reformation as ULF waves impinge upon it. We acknowledge that there may be additional ion effects at smaller kinetic scales, pending further study and resource expenditure. A convergence study would be a very

expensive undertaking, but we intend to further analyse these effects in future studies. To our knowledge, there doesn't exist a study yet which would invalidate the dynamics seen with this resolution.

***It is not clear why authors have chosen to do the test particle approach if the Vlasov code is supposed to follow the full distribution function. This point needs to be clearly explained.***

On line 210-213 we state: "Following the evolution of distribution functions does not allow for tracing of particle histories. In order to evaluate injection probabilities, particles need to be tracked as they meet the bow shock and interact with it, ultimately either returning to the upstream or being transmitted to the downstream." We will reword this for added clarity, highlighting how we actually use these test-particles as a method of tracking the evolution of a small portion of the VDF.

***The concept of non-locality is introduced which does not include magnetic field. The motivation is that it provides poor results while at the same time paper mentions that magnetic field structures are very important for the injection. All this makes the motivation for the non-locality concept very unsatisfactory and it is not clear what authors mean motivating their selection by having poor vs good results. Magnetic field data is one of the primary datasets in the shock analysis and it is unclear why one would want to exclude it from the shock definition. In general, it is not clear why authors want to introduce a new concept.***

We acknowledge that magnetic field measurements are often used in spacecraft. We will clarify this section, stating that (in agreement with analytical studies of quasi-parallel shocks resulting in little magnetic field compression), the magnitude of magnetic field at the quasi-parallel shock showed multiple successive enhancements and rarefications. As our proposed method depended upon the measurement reaching a conclusive downstream state, the magnetic field magnitude as such was insufficient. We do, however note that the magnetic field does have a role in the calculation of the
shock-normal magnetosonic Mach number, so the magnetic field is not ignored.

*In Figure 4 100eV case of test particles is shown. It should be motivated why this particular case is shown and not for example the case of Maxwellian distributed particles.*

The Maxwellian case appears very similar to the presented 100 eV case, so we shall replace the figure. We also note that the evolution of all performed test-particle distributions can be examined in supplementary movies B and C.

*Figure 5 results and discussion are not fully consistent and should be significantly improved. For example, showing injected particle results (column 1 and 2) one makes conclusion that particles with energies below solar wind drift energy are loosing energy on average and particles above are gaining. This results is inconsistent with that the figures shows most of the injected particles have high energy. Such high energy particles if they start at solar wind energy and then during some part of the orbit have energies below the solar wind energy then on average at low energies the energy gain and loss should be equal. If the statement made in the manuscript is true then why there are no low energy injected particles (while there are still a lot of low energy particles at r<0 and they all show negative energy gain.*

We wish to thank the referee for pointing out this error. We did additional analysis of our results, and found that when binning the particle energization changes, the script made an erroneous assumption that energy changes per time step would be very small. Due to each energy change being recorded at the end value, the plot emphasized energy gains at high energies and losses at low energies.

We are in the process of fixing our analysis and completely redoing figure 5 and the associated interpretation.

*Similarly, it is not clear how the current simulations results contradict the re-*

*sults from the Johlander et al. Firstly, it is not clear if SLAMS are observed in the current simulation and if they are do they have similar properties as in the observations? Secondly, when comparing with Johlander et al., it would be good to do the comparison in an adequate way, so that one understand how one should translate the results from the Vlasiator case to another cases such as Johlander et al. For comparison with those results one would need to look at solar wind ions that have different kinetic energy in the shock frame and see the differences in the injection rate. The authors should guide the reader where and how this can be seen.*

We acknowledge that the comparison was very brief. In Johlander et al, Fig. 5, low-energy particles were found to be likely to reflect from SLAMS, whereas fast particles passed through them. In the quasi-parallel region, structures such as SLAMS merge into the bow shock, and thus, there is some merit in comparing how particles interact with SLAMS vs how they interact with the bow shock. Our results showed that particles with a large energy in the solar wind frame were more likely to be injected, but we acknowledge that a high solar wind frame speed can result in both faster and significantly slower shock-frame particles. We will perform additional tests based on the shock-frame energy and revise the results accordingly.

*Figure 6 requires several clarifications. The largest structuring of the injection probabilities is seen in the dependance on the impact position angle. Instead of trying to resolve the physics of the large injection rate variations authors suggest how to smooth these variations which suggests that authors themselves maybe do not trust the numbers. This needs to be clarified.*

We indeed expect that panel to not depict any underlying erratic dependence on shock-normal angle, but rather, to be indicative of how our particle injection time window was not long enough to encompass a sufficient amount of shock reformation cycles. We will clarify this reasoning. Due to the large spatial extent and the large amount of test-particles, we still believe our statistics are sufficient to probe other properties of

particle-shock-interactions.

In an ideal world, we would use a longer period of time for our test-particle study, but such a simulation set is not available at this time.

***Another unclear point is how shock non-locality is defined for a particle that starts at one position and gets injected at another position (in general valid for all particles). From which time and position are the given shock non-locality values. Similarly, it is not clear at which time instant is measured the bow-normal angle.***

We acknowledge that the statement on lines 289-290: "For each test-particle, we evaluate these properties at the first time the particle reaches a point in the simulation space that fulfills the solar wind core heating ($T_{core} > 4T_{sw}$) criterion." was somewhat hidden. We shall clarify this point in the text and the caption.

***Minor things:***
***L.8 fix the language of the sentence.***

Thank you for this correction.

***l.45 The work of Johlander et al. 2016 does not make the mentioned assumptions in the manuscript but shows that SLAMS can contribute to the injection.***

The referee is correct, the assumptions of the test-particle study in Johlander et al (2016) differ in that they investigate a SLAMS instead of a planar shock front. We will correct this mistake.

***l.59 It is a bit confusing in which reference frame particle gains energy in the definition of the energization. For example, a particle reflected from a shock can have lower energy in the shock reference frame than the solar wind particles (e.g. when reflected from SLAMS) but it would not be "part of the incident thermal distribution".***

We thank the referee for making this important point, and we will add it to the

manuscript.

**I.165 the division of the core distribution is unclearly described.**

We will reword this to the following: "The Vlasiator distribution function is separated into core and suprathermal parts ($n_{p,core}$ and $n_{p,st}$). Each velocity space cell is evaluated as belonging to the core distribution, if it is inside a sphere centred at $u_{sw} = (-600, 0, 0)$ km/s and with a radius of 690 km/s. Cells outside this sphere are considered as belonging to the suprathermal distribution."

**I.368 this should be illustrated and quantified, adding by the figure**

We will reword this for clarity – this in fact was referring exactly to Figure 5, which shows changes in particle energy as measured in the simulation frame. We will also amend the caption of Figure to this effect, and if necessary, change the text according to our re-done analysis.

**I.411 What do you mean by "high-fidelity"?**

We will replace "high-fidelity" with "noise-free".

**Figure 1: please use slightly thinner lines, the structure of the pink line cannot be resolved in the figure due to the thickness of the line.'**

We did not intend Figure 1 to be used for evaluating the mesoscale bow shock shape, instead highlighting this in Figure 2. Nevertheless, we will redo Figure 1 with a smaller line width to show the details already in this image.

**Table 1: Why comparison is done with suprathermal densities? I assume that from Vlasiator one can estimate the flux of reflected particles and thus have a good estimate of the injection rate.**

Within the foreshock region, the suprathermal density indeed is a close measure of reflected particles, with only possible very minor contamination from the solar wind core during flow deflection events. The datasets of Vlasiator are too large (several terabytes)

to store all data at every time step, thus we use reduced measurements such as the described split into core and suprathermal portions of the distribution function. We have added a description to explain that the suprathermal density is a good measure of reflected particles, adding the note that it includes all particles which have been reflected, even a long time ago, and are currently in the upstream.

---

## Author Response (AR2)

**Dear esteemed editor and referees,**
**We the authors thank you for the assessment of our manuscript. We have made changes requested by referee #2 and discuss the points below, with the referee comments in italics and our responses in boldface.**

*While the non-locality and it's relation to the injection is in the title of the paper, there is very little major conclusions in the paper about this connection. For example, lines 378-285 discussing this relation are very weak, there is no efforts for example to check the statement "it may be related to a particularly strong local magnetic field twist or some other transient ". This is definitely a weakness of the paper, because if there is no strong conclusions on the importance of non-locality, the introduced term "non-locality" looses it's attraction. Reading the abstract the impression is that non-locality is not important and then there is a question, why it should be in the title?*

**We thank the referee for helping us improve the paper in this respect. We have reworked the abstract and adjusted the discussion and conclusions to better evaluate the significance of non-locality. We believe the title of the manuscript should remain mostly as is, but would be happy to adjust it to**
**"Non-locality of the Earth's quasi-parallel bow shock and injection of thermal protons in a hybrid-Vlasov simulation" if this would be preferable. Replacing the colon with "and" might clarify that we discuss both topics separately and investigate the link between them.**

*l.91-99 What is conclusion from Toth et al. study? Is it ok to ignore the kinetic scales? Somehow after the reading the paragraph it is still not clear why underresolving the kinetic scales is ok to address the topic of the paper. What does it mean "correct scale separation between global and local dynamics"? What are "the number of ion kinetic effects"? Just writing down more specific what one is able and not able to do and which part of the injection problem one is addressing would help more readers.*

**Sadly, the parameter range study of Toth et al. did not extend to underresolving the ion inertial length. In the MHD-EPIC runs they tested $d_i/\Delta x$ values ranging between 10 and 40 (where $d_i$ is the ion inertial length and $\Delta x$ is the spatial grid resolution), and in Hall-MHD runs between 5 and 20. Thus, they did not prove that it is safe, nor did they prove it to be incorrect either. We believe there is much interest in investigating this topic and hope to approach it soon, but it is a study in its own right and requires new large-scale simulation runs to perform. We of course would be very interested in reading any existing literature on this topic, but have not found such papers so far.**

**The scale separation refers to $d_i/d_g$, where $d_g$ in Toth et al is a global system size which they characterize in their simulations as the magnetopause standoff distance of about 10 $R_E$. Near dayside reconnection they state the true proton inertial length as $d_p{\sim}0.01\ R_E$ or 63 km. They state correctly that $d_i/d_g$ needs to be a small number, but in their chapter 5 they show how they scale $d_i$ up as $d_i=f*d_p$ starting from a value of $f=8$ and going up to values of $f=128$. We agree that a value of 8 is still such that global and inertial scales are separated ($d_i/d_g{\sim}0.008$) but at $f=128$ they already have $d_i>R_E$ where scale separation is no longer true in our opinion. We would in fact argue that the important global scale should**

be much smaller than the magnetopause standoff distance, and closer to one Earth radius. For example, magnetopause flux transfer events and magnetosheath pressure pulses are going to cause disturbances on scales closer to a few than then $R_E$. Additionally, scaling the ion inertial length up by a factor of 8 should also increase the size of any related structures by a factor of 8, altering how they might interact with other mesoscale phenomena such as ULF waves.

We also note that if we would use a scaling such as they presented and artificially increase our inertial length by e.g. a factor of 8 to 504 km, our grid resolution would resolve the ion inertial length, but to achieve this our ion mass would need to increase to $64*m_p$. In order to maintain the validity of acceleration effects, the charge would also need to increase similarly. In order to maintain solar wind mass loading, the number density would need to be scaled down. The result would be that both ion gyro- and plasma frequencies would in fact not change, and thus the inertial length could not be scaled up without relaxing some of our validity requirements such as the charge-to-mass ratio (which Toth et al indeed relaxed).

Since we do not perform re-scaling of ion masses, we have not included the discussion above in the manuscript itself. Our team does have a manuscript nearing submission which proves that using current resolutions, Vlasiator resolves and models well EMIC waves and mirror modes in the magnetosheath, and thus these kinetic effects at least are within the simulatable domain. As Annales Geophysicae accepts references to manuscripts under preparation, we have added a short reference to this study (Dubart et al 2020). We also have added a reference about Vlasiator modelling global reconnection rates in agreement with empirical formulae (Hoilijoki et al 2017). We also added an example of scale separation to help clarify the topic.

*Figure 5:*
*I am very confused about the transmitted particle plot. What does it mean that at high energies the energy decrease is negative only? How particles can arrive to that region in the first place if they start at low energies?*

This is the section where the request from the previous round helped us to reformulate the results. In the previous round, we added the sentence "As energy gains and losses can be significant near strong electric fields (up to 1 keV per measurement interval), we use the initial energy of each change as the y-coordinate" on line 282. Thus, in the top half of the panels, it is not that there is only energy decrease, but that of all energy changes originating in that part of the plot, the average is in the energy decrease direction.

We have improved the Figure 5 caption to clarify these choices and their significance, as well as improve the body of the text to the same effect.

*I am also confused about 5e and 5g plots. Upstream in the solar wind we should have mainly 10eV particles but the logarithmic count statistics lines show few particles around those energies. This is particularly clear in 5g. How am I supposed to interpret this?*

There are a number of effects at play here, which are jointly responsible for the apparent effect. The most significant reason is that 10 eV particles in the upstream are least likely to spend much time around 0.9 $R_E$ (as they are rapidly convected along with the plasma flow) whereas higher energy ~100 eV particles can gyrate more and cause more particle counts to be registered upstream of the estimated shock position, while still finally ending up in the downstream and as transmitted particles. This is visible in panels c and g.

Evaluating the solid thin contour in panel e) and the dashed thick contours in panel g) shows somewhat enhanced counts extending to below 10 eV whereas in panels f) and h) for the 100 eV initialization energy the contours show that significant counts do not extend to such low energies. Thus, we conclude that we do not suspect any error in the initialization energy of those particles. We have amended the manuscript to clarify this effect.

*I am also surprised that there is no increasing heating at the shock itself (dx=0).*

Indeed, energization does not seem to happen point-like at the shock dx=0, but rather over a larger region which starts at roughly dx=0 and extends to the downstream. We have emphasized this result in our amendments to the manuscript.

*The maximum penetration depth downstream (1.2$R_E$) is relatively close to the maximum non-locality values (1.2$R_E$). Can it be that it is not penetration downstream but that those are the cases with highest non-locality measure and thus the cases where the shock position can be at negative dx values.*

We selected the $\Delta x(T_{core})$ metric because our visual estimates say it is the most stable of the three selected shock parametrizations. However, the referee is correct that in some cases non-locality could indeed cause this. For example, in Figure 4f at about Y=3 $R_E$ there is a large non-locality value and energetic injecting particles might indeed be registered as being far behind the shock. We have added this discussion to the manuscript and thank the referee for pointing this out.

*l.258 Please specify the Maxwellian temperatures in eV in the paper. It is easier comparison with the other data in the manuscript where energy is measured in eV.*

For a 0.5 MK Maxwellian distribution the mean square speed is 112 km/s and most probable speed is 91 km/s. These values correspond with energies of 65 eV and 43 eV, respectively. We added this to the final paragraph of section 4.

[revised manuscript text omitted]